# Climate & Ecology in the Rocky Mountain Interior After the Early Eocene Climatic Optimum

Rebekah A. Stein[1,2], Nathan D. Sheldon[1], Sarah E. Allen[3], Michael E. Smith[4], Rebecca M. Dzombak[1], Brian R. Jicha[5]

[1]Department of Earth & Environmental Sciences, University of Michigan, Ann Arbor, MI, 48109, USA
[2]Department of Earth & Planetary Sciences, University of California, Berkeley, CA, 94720, USA
[3]Department of Biology, Penn State Altoona, Altoona, PA, 16601
[4]School of Earth & Sustainability, Northern Arizona University, Flagstaff, AZ, 86011, USA
[5]Department of Geosciences, University of Wisconsin-Madison, Madison, WI 53706

*Correspondence to*: Rebekah A. Stein (restein@berkeley.edu), Nathan D. Sheldon (nsheldon@umich.edu)

**Abstract.** As atmospheric carbon dioxide ($CO_2$) and temperatures increase with modern climate change, ancient hothouse periods become a focal point for understanding ecosystem function under similar conditions The early Eocene exhibited high temperatures, high $CO_2$ levels, and similar tectonic plate configuration to today, so it has been invoked as an analog to modern climate change. During the early Eocene, the greater Green River Basin (GGRB) of southwest Wyoming was covered by an ancient hypersaline lake (Lake Gosiute; Green River Formation) and associated fluvial and floodplain systems (Wasatch and Bridger Formations). The volcaniclastic Bridger Formation was deposited by an inland delta that drained from the northwest into freshwater Lake Gosiute and is known for its vast paleontological assemblages. Using this well-preserved basin deposited during a period of tectonic and paleoclimatic interest, we employ multiple proxies to study trends in provenance, parent material, weathering and climate throughout one million years. The Blue Rim escarpment exposes approximately 100 meters of the lower Bridger Formation, which includes plant and mammal fossils, solitary paleosol profiles and organic remains suitable for geochemical analyses, as well as ash beds and volcaniclastic sandstone beds suitable for radioisotopic dating. New $^{40}Ar/^{39}Ar$ ages from the middle and top of the Blue Rim escarpment constrain age of its strata to ~49.5–48.5 Ma ago, during the "falling limb" of the early Eocene climatic optimum. We used several geochemical tools to study provenance and parent material in both the paleosols and the associated sediments and found no change in sediment input source despite significant variation in sedimentary facies and organic carbon burial. We also reconstructed environmental conditions, including temperature and precipitation (from paleosols) and the isotopic composition of atmospheric $CO_2$ from plants found in the floral assemblages. Results from paleosol-based reconstructions were compared to semi-co-temporal reconstructions made using

leaf physiognomic techniques and marine proxies. The paleosol-based reconstructions (near the base of the section) of precipitation (608–1167 mm yr$^{-1}$) and temperature (10.4 to 12.0 °C) were within error of, although lower than, those based on floral assemblages, which were stratigraphically higher in the section and represented a highly preserved event later in time. Geochemistry and detrital feldspar geochronology indicate a consistent provenance for Blue Rim sediments, sourcing predominantly from the Idaho paleoriver, which drained the active Challis volcanic field. Thus, because there was neither significant climatic change nor significant provenance change, variation in sedimentary facies and organic carbon burial likely reflected localized geomorphic controls, and the relative height of the water table. The ecosystem can be characterized as a wet, subtropical-like forest (i.e., paratropical) throughout the interval based upon the floral humidity province and Holdridge life zone schemes. Given the mid-paleolatitude position of the Blue Rim Escarpment, those results are consistent with marine proxies that indicate that globally warm climatic conditions continued beyond the peak warm conditions of the early Eocene climatic optimum. The reconstructed atmospheric $\delta^{13}C$ value (−5.3 to −5.8‰) closely matches both the independently reconstructed value from marine microfossils (−5.4‰), which provides confidence in this reconstruction. Likewise, the isotopic composition reconstructed matches the mantle most closely (−5.4‰), agreeing with other postulations that warming was maintained by volcanic outgassing rather than a much more isotopically depleted source, such as methane hydrates.

# 1 Introduction

## 1.1 The Eocene Period as an Analog for a Future Warm World

The anthropogenic release of fossil fuels drives a rapid and sustained increase in atmospheric carbon dioxide ($CO_2$) that is coupled with climate change (IPCC 2007). To understand the effects of elevated $CO_2$ on the Earth (e.g., Cotton et al., 2013), we seek out geological periods with high temperatures and high atmospheric $CO_2$ for comparison. The early Eocene climatic optimum (EECO) has been invoked as a climate analog for our projected future (e.g., Zhu et al., 2019). This warming during the EECO occurred 53.26–49.14 million years ago (Cramwinckel et al., 2018; Westerhold et al., 2018), with peak warming from 51.5–50.9 Ma; this period consisted of long-term global temperature maxima and high $CO_2$ levels but was tectonically comparable to today (Hyland & Sheldon 2013; West et al., 2020). From the Paleocene to early Eocene, it has been inferred that there were extensive temperate forests dispersed throughout North America (Smith et al., 2012; Breedlovestrout

et al., 2013; Greenwood et al., 2016; West et al., 2020) up to high latitudes 65 °N (Dillhoff et al., 2013). However, the nearby Bighorn Basin is inferred to have undergone aridification based on magnetic properties in paleosols (Maxbauer et al. 2016; Carmichael et al. 2017), and global climate models predict low and lower-middle latitude sites, including areas like central Utah to experience aridification due to changes in meridional vapor transport distribution (Pagani et al., 2006). As the planet warms, there is increasing concern about water availability and dry climates getting drier. For example, the North American Southwest, composed of a series of deserts and dry ecosystems, is at risk for having its already severe droughts increased in frequency and severity (Poore et al., 2005; Coats et al., 2015; Cheeseman 2016). Therefore, the study of ancient climate and ecosystems in these hydrologically vulnerable areas can provide examples for what may happen to these ecosystems in the context of emerging climate and societal challenges.

## 1.2 Continental interior and foreland basins in the Rocky Mountains

The marine foreland of the North American Cordillera was partitioned into a series of terrestrial basins by an anastomosing network of Laramide basement structures during the Paleogene, likely due to coupling between the shallow Farallon slab with the North American lithosphere (Dickinson et al., 1988; Snoke et al., 1993; DeCelles, 2004). Geomorphic evolution of drainage patterns during the Paleocene and Eocene that were driven by Laramide orogenesis led to the formation of a series of large lakes, preserved as several kilometers of lacustrine and fluvial strata (Smith et al., 2008). High sediment accumulation rates in these basins contributed locally to excellent preservation of the biota, allowing us to study deep time at high resolution (Looy et al., 2014). The carbonate-rich lacustrine Green River Formation was deposited from Lake Gosiute within the greater Green River Basin (Fig. 1). The predominantly siliciclastic Wasatch and Bridger formations reflect contemporaneous fluvial and floodplain deposition adjacent to Lake Gosiute.

The greater Green River Basin lies to the east of the Sevier fold and thrust belt, and is bounded on the north, east and south by Laramide basement structures, each of which likely contributed water and sediment to the basin (Smith et al., 2008; Smith et al., 2014). The basin also likely received water and sediment at times from paleoriver(s) which drained the high-elevation Cordilleran hinterland to the west (Chetel & Carroll, 2010; Chetel et al., 2011; Smith et al., 2014). Over two kilometers of terrestrial strata accumulated in it depocenter during the Paleogene. The paleolatitude of southwest Wyoming in

the early Eocene (~41 °N; Wolfe et al., 1998; Hyland & Sheldon 2013) is thought to be relatively comparable to its position today, thus changes in the climate of this region are not related to changes in latitude.

### 1.3 Using multiple proxies to characterize an environment

High resolution, well-informed snapshots in time of individual regions with thorough climate reconstructions help us to understand climate change and subsequent ecosystem dynamics (Guiot et al., 2009; Li et al., 2010; Shala et al., 2017). There are several terrestrial proxies that can be used to reconstruct paleoclimate and paleoecology, e.g., pedogenic carbonate isotope

values (Cerling 1992), floral assemblages (Wilf 2000; Spicer et al., 2009), and stomatal density (Royer 1999). The preservation of abundant, high-quality organic and inorganic specimens and samples due to the tectonic assemblage of the basin, makes the GGRB an excellent location for a multi-proxy approach. Organic geochemistry, specifically, isotopic values of plants ($\delta^{13}C_{plant}$) have been used to reconstruct the value of the isotopic composition of the atmosphere – termed $\delta^{13}C_{atm}$ (Arens et al., 2000; Stein et al., 2019). $\delta^{13}C_{atm}$ reflects sources of $CO_2$ gas to the atmosphere (Keeling 1979; Boutton 1991); for example, mantle

degassed values of $\delta^{13}C_{atm}$ tend to be around −5.4 ‰ (Deines 1992). Identifiable organic fossils with individually compressed leaves can be used to reconstruct the value of the atmosphere. Multiple inorganic geochemistry proxies (see methods) can provide context for the depositional environment (e.g., Sheldon et al., 2006), climate (e.g., Gallagher & Sheldon 2013), hydroclimate (e.g., Sheldon et al., 2002), age (e.g., Turner 1971), and origin of sediments (e.g., Sheldon & Tabor 2009). Together, these proxies can inform us on ecosystem and depositional dynamics in the context of Eocene climate. This study

seeks to combine these proxies to create a holistic reconstruction of the greater Green River Basin (GGRB) during the early Eocene using the extensive deposits of the Blue Rim escarpment.

### 2 Description of locality

The Bridger Formation is an approximately 842 m thick series of tuffaceous deltaic-alluvial and minor lacustrine

sedimentary strata which overlie and interfinger with the Green River Formation (Koenig 1960; Kistner 1973; Murphey & Evanoff 2001; Clyde et al., 2001). Vertebrate fossils collected from the Bridger Formation formed the basis for defining the Bridgerian North American Land Mammal 'age' ("NALMA"; e.g., Osborn 1909; Wood et al., 1941; Van Houten 1944;

Gingerich et al., 2003; Robinson et al., 2004). Mapping and biostratigraphy have permitted further subdivision of the Bridger Formation into intervals A-E (Matthew 1909; Murphey & Evanoff 2001; Murphey et al., 2011). Several volcanic ash horizons within the Bridger Formation and the underlying Green River Formation have been radioisotopically dated using $^{40}Ar/^{39}Ar$ geochronology to have accumulated between approximately 50 Ma and 46 Ma (Fig. 1; Table S2; Murphey & Evanoff 2001; Smith et al., 2008). Radioisotopic ages reported or discussed in this contribution have all been calculated using the 28.201 Ma age for the Fish Canyon tuff sanidine standard, and are thus comparable with modern U-Pb geochronology (Kuiper et al., 2008; Smith et al., 2010). Strata exposed along the Blue Rim study area represent the oldest exposed portion of the Bridger Formation. Unlike much of the Bridger Formation, these 'Bridger A' deposits have not been radioisotopically dated, but regional mapping and correlations suggests they were deposited above the Sand Butte bed of the Laney Member of the Green River Formation, and likely occur beneath the more extensively mapped Bridger B interval (Murphey & Evanoff 2001), bracketing a depositional age between the ca. 50 Ma '6th tuff' of the Green River Formation and the ca. 49 Ma age for the Church Butte tuff, which occurs within Bridger B (Fig. 1).

Several potential sediment sources may have contributed to the Bridger Formation: including siliciclastic material of a variety of compositions derived from Phanerozoic strata and underlying basement exposed by Laramide uplifts which surround the basin (Smith et al., 2015); volcaniclastic and siliciclastic sediment delivered by the Idaho paleoriver (Chetel et al., 2011); volcanic ashfall from the Challis and Absaroka volcanic fields; and autochthonous lacustrine carbonates (Murphey & Evanoff 2001; Fig 1a). Evidence supporting these as potential sources include common detrital feldspar ages in the sand grains that are similar to depositional ages (i.e., recently erupted volcanic grains), and many euhedral volcanic biotite and felsic volcanic lithic grains in the Bridger Formation sandstones (Chetel et al., 2011). Whereas the older, and in part coeval Laney Member of Green River Formation is composed primarily of carbonate lacustrine sediments, the Bridger Formation is composed principally of siliciclastic sediment, with several minor intervals of lacustrine carbonate (Buchheim et al., 2000; Murphey & Evanoff 2001). Sediment accumulation during Bridger Formation deposition appears to have been relatively continuous in the basin center based on existing age control (Murphey & Evanoff 2001). Because of this, the Bridger Formation has pristine faunal and floral preservation, making it an excellent candidate for understanding ecosystem function (Brand et al., 2000; Allen et al., 2015; Allen 2017b).

The Bridger Formation is exposed laterally over 12 kilometers at Blue Rim and is locally approximately 100 m thick (Kistner 1973). The flora at Blue Rim has been collected and described in great depth and is known for excellent plant

preservation including leaves, flowers, fruits, seeds, wood, pollen, and spores (Allen 2017a/b). Eocene floral assemblages at Blue Rim occupy warm, wet biomes not unlike modern subtropical ecosystems; angiosperms including palms were abundant and dicotyledonous taxa were up to 28 m tall (Allen 2015; Allen 2017a/b), antithetical to the dry scrub desert found at Blue Rim escarpment today. Of the multiple quarries created for plant fossil excavation (see Allen 2017a/b), the lower horizon (older) floral assemblage consists of taxa such as the abundant climbing fern *Lygodium kaulfussi* (fern, Lygodiaceae)*,* "dicots"

like "*Serjania" rara* (soapberry, Sapindaceae)*, Populus cinnamomoides* (poplar, Salicaceae) and *Landeenia arailiodes* (Sapindales *, Goweria bluerimensis* (Icacinaceae)*,* and *Phoenix windmillis* (palm; Arecaceae; Allen 2015; Allen et al., 2015; Allen 2017b)*.* The upper horizon preserves taxa such as *Macginitiea wyomingensis* (sycamore; Platanaceae)*, Populus cinnamomoides* (poplar, Salicaceae)*, Cedrelospermum nervosum* (elm; Ulmaceae)*, "Serjania" rara* (soapberry, Sapindaceae), and many more (Allen 2017b).

**3 Methods**

**3.1 Geochronology**

Volcaniclastic beds were sampled from the Blue Rim escarpment for $^{40}Ar/^{39}Ar$ geochronological dating (e.g., Turner 1971): two samples from the prominent 'blue-green marker' unit (Fig. 3, 31 m in Figs. 4—7) that occurs approximately halfway

up the section; and two sand beds that crop out near the top of the exposure. To prepare sanidine phenocrysts for analysis, samples were crushed, leached in dilute hydrochloric acid (HCl) and hydrofluoric acid (HF) prior to hand-picking of sanidine in refractive index oils using a petrographic microscope, and then ultrasonic cleaning in acetone and ethanol. Sanidine phenocrysts from sampled ash beds were irradiated adjacent to standard sanidine crystals from Fish Canyon tuff (FCs) in cadmium shielding within the TRIGA (Training, Research, Isotopes, General Atomics) water-cooled, low-enriched

uranium/zirconium fuel reactor at Oregon State University. Single sanidine crystals were fused using a 25W $CO_2$ laser and then analyzed for $^{40}Ar/^{39}Ar$ composition using a MAP 215-50 mass spectrometer attached to a metal ultra-high vacuum (UHV) gas extraction and clean-up line at the University of Wisconsin Madison WiscAr laboratory. A 28.201 Ma age for Fish Canyon

sanidine standard (FCs; Kuiper et al., 2008) was used to calculate apparent ages for each laser fusion analysis, and weighted mean ages were calculated for the youngest coherent population of sanidine dates from each sample. For populations that exceed an MSWD of 1, uncertainties in the weighted mean were multiplied by the inverse of the square root of the MSWD to reflect the additional uncertainty implied by the associated age scatter.

## 3.2 Physical Measurements

### 3.2.1 Stratigraphy and fossils

Starting at the base of the Blue Rim escarpment, adjacent to the first sampled paleosol (Fig. 2), an updated stratigraphic column to that found in Allen's (2017b) dissertation was measured (41.7985 °N, -109.5856 °E) in September 2019 (Fig. 3, 4, 5, 6, 7). This 67 m stratigraphic column traced the flanks of the escarpment to the top of the badlands. The stratigraphic column was sampled every 3 m (approximately the height of two Jacob-staffs) or at every interval of visual change (color, texture). In addition, plant fossils and plant hash were quarried at two locations approximately halfway (26 m, 33 m) and close to the top of the stratigraphic section (51.5 m, 52 m) for organic-rich fossil samples for isotope and bulk chemistry analysis.

### 3.2.2 Paleosol sampling

Six profiles of the same paleosol were identified and excavated along a lateral transect at the base of the Blue Rim escarpment (41.79892625, -109.58362614, WGS 1984 (19BRWY1); Fig. 7, S8). Paleosols sampled by horizon based on pedogenic features including root traces, burrows, drab-haloed and kerogenized roots, and horizonation (Fig. S1a, Table S1). Fresh rock material was excavated by digging at least 20 cm into the surface, avoiding all traces of modern pedogenesis or surficial climate influence (i.e., modern roots or carbonate nodules). One distinctive, laterally continuous paleosol at the base of the section was sampled in five individual profiles over 440 m (Fig. 3; Fig. S1, S2, 19BRWY2-6). Samples were collected from each horizon present, with no fewer than three samples per paleosol profile. At each location, every present horizon (A, B, C) was sampled. Epipedons were present for all paleosols except 19BRWY4. For horizons that had color, texture, and/or other physical intra-horizonal changes, multiple samples were collected.

### 3.2.3 Isotope analyses

Paleosol samples were ground to 70 μm in a shatterbox. Approximately 10 g aliquots of samples were weighed out and then acidified in 5% hydrochloric acid (HCl) for 30 minutes to remove carbonate and leave behind total organic carbon of bulk sample. After 30 minutes, these samples were decanted, then re-acidified for a total of three times (and/or until solution stopped bubbling). After these acid washes, they were rinsed with deionized water three times (or more, if given >3 acid washes). Samples were then dried in an oven at 50 °C for 72 hours.

Between 20–25 mg of each sample was loaded into tin capsules and run on a Costech Elemental Analyzer to determine weight %C and %N in University of Michigan's Earth Systems Lab with acetanilide (71.09 %C, 10.36 %N) for elemental composition calibration and acetanilide and atropine (70.56 %C, 4.18 %N) standards. The %C was used to calculate idealized loading size for isotope analysis; these samples were then run on the Picarro Cavity Ring-Down Spectroscopy (CRDS) on low carbon (Mode 9) for organic carbon isotope values ($\delta^{13}C_{org}$), with external precision better than $\pm 0.3$‰ for low-carbon samples,

and internal standard replication of <0.34‰ for all standards.    Specimen aliquots weighing 20–25 mg were run with IAEA standards (IAEA-CH6: Sucrose, $\delta^{13}C$ = -10.45‰; IAEA-600: Caffeine, $\delta^{13}C$ = -27.77‰) and laboratory internal standards ($C_3$ sugar, $\delta^{13}C$ = -26.18‰, $C_4$ sugar, $\delta^{13}C$ = -12.71‰, acetanilide, $\delta^{13}C$ = -28.17‰).

### 3.2.4 Bulk geochemistry

Approximately 10 g aliquots of crushed paleosols, mudstone, and siltstone were measured and sent to ALS Laboratories in Vancouver, British Columbia, Canada for bulk elemental analysis. At ALS, samples were digested with perchloric ($HClO_4$), hydrofluoric (HF), nitric ($HNO_3$) and hydrochloric (HCl) acids, and concentrations were determined by inductively coupled plasma (ICP) optical emission spectrometry and ICP-mass spectrometry. The ICP-OES and ICP-MS were calibrated using internal standards, with major element precision better than 0.2 weight %, and error was calculated based on

maximum error of duplicates and standard tolerance. Additional information regarding methodology is proprietary, and can be sought from ALS Laboratories directly. Elements measured included Aluminum (Al), Arsenic (As), Barium (Ba), Calcium (Ca), Iron (Fe), Potassium (K), Lanthanum (La), Magnesium (Mg), Sodium (Na), Sulfur (S), Titanium (Ti) (weight %) and

Silver (Ag), Beryllium (Be), Bismuth (Bi), Cadmium (Cd), Cerium (Ce), Cobalt (Co), Chromium (Cr), Cesium (Cs), Copper (Cu), Gallium (Ga), Germanium (Ge), Hafnium (Hf), Indium (In), Manganese (Mn), Molybdenum (Mo), Niobium (Nb),

Phosphorus (P), Lead (Pb), Rubidium (Rb), Rhenium (Re), Antimony (Sb), Scandium (Sc), Selenium (Se), Strontium (Sr), Tantalum (Ta), Tellurium (Te), Thorium (Th), Thallium (Tl), Uranium (U), Vanadium (V), Tungsten (W), Yttrium (Y), Zinc (Zn), Zirconium (Zr) (weight ppm).

## 3.3 Paleoclimatic and Paleoenvironmental Calculations

### 3.3.1 Weathering indices and leaching

Weathering was quantified using Chemical Index of Alteration of B horizons (CIA; Equation 1; Nesbitt & Young 1982), a feldspar weathering index based on the discrepancy in ion mobility during weathering.

$$CIA = \frac{Al_2O_3}{Al_2O_3 + Na_2O + CaO + K_2O} \times 100 \tag{1}$$

To test for alteration and expected pedogenic elemental trends, changes in individual element mobility and strain were explored using mass balance (Equations 2–3; Chadwick et al., 1990), where $\varepsilon$ represents the strain on an immobile element like Ti or Zr and $\tau$ represents the relative gain or loss of a mobile element relative to the paleosol's parent material. Average values of 2.7 g cm$^{-3}$ and 1.5 g cm$^{-3}$ were used for reworked ash and soil density, respectively (e.g., Sheldon & Tabor, 2009).

$$\tau_{j,w} = \frac{(\rho_w C_{j,w})}{(\rho_p C_{j,p})} \times [\varepsilon_{i,w} + 1] - 1 \tag{2}$$

$$\varepsilon_{i,w} = \frac{(\rho_p C_{j,p})}{(\rho_w C_{j,w})} - 1 \tag{3}$$

The Paleosol Weathering Index (PWI; Equation 4; Gallagher & Sheldon 2013), which is based on differential bond strengths in cation oxides, provided additional means for examining weathering. Molar concentrations are used to make

calculations in Equations 1–4, rather than elemental concentrations.

$$PWI = ((4.20 * Na) + (1.66 * Mg) + (5.54 * K) + (2.05 * Ca)) * 100 \qquad (4)$$

We used several geochemical proxies to examine intensity of leaching, including the ratio of barium to strontium (Ba/Sr), which is higher with more leaching and lower with less leaching (Sheldon 2006; Retallack 2001) due to differential solubility; Sr is more soluble than Ba (Vinogradov 1959). The ratio of the sum of base cations to titanium is another metric for leaching, under the assumption that titanium is immobile, while other bases are mobile (Sheldon & Tabor 2009). The ratio of the sum of base cations to aluminum has been used as a metric for hydrolysis (Retallack 1999; Bestland 2000; Sayyed & Hundekari 2006).

### 3.3.2 Provenance and Parent Material

The molar ratios of titanium to aluminum (Ti/Al) and zirconium to aluminum (Zr/Al) was used to screen for consistency in sediment source in soils; direction of change in Ti/Al ratios is related to differences in chemical weathering, while Zr/Al ratios are related to changes in physical weathering (Sheldon & Tabor 2009). The molar ratios of uranium to thorium (U/Th), and lanthanum to cerium (La/Ce) were used to trace potential changes in parent material composition through the stratigraphic unit, where a constant down-profile U/Th and La/Ce ratios reflect single-parent source (Sheldon 2006; Sheldon & Tabor 2009). Absolute parent material values for each of these ratios are not well calibrated, but direction of change observed at any site indicates a change in parent material. The U/Th ratio is redox-sensitive, so La/Ce ratios are used as a comparative point in case of highly-reduced environments, resulting in skewed U/Th ratios.

The molar ratios of titanium to aluminum (Ti/Al) and zirconium to aluminum (Zr/Al) were used to screen for consistency in sediment source in soils; direction of change in Ti/Al ratios is related to differences in chemical weathering, while Zr/Al ratios is related to changes in physical weathering (Sheldon & Tabor 2009). The molar ratios of uranium to thorium (U/Th), and lanthanum to cerium (La/Ce) were used to trace potential changes in parent material composition through the stratigraphic unit, where a constant down-profile U/Th and La/Ce ratios reflect single-parent source (Sheldon 2006; Sheldon & Tabor 2009). Absolute parent material values for each of these ratios are not well-calibrated, but direction of change observed at any site indicates a change in parent material, U/Th is redox-sensitive, so La/Ce ratios are used as a comparative point in case of highly reduced environments.

### 3.3.3 Paleoclimate reconstructions using foliar assemblages

Before and up to 2017, co-author SAE surveyed and described 69 leaf morphotypes collected from multiple quarries at Blue Rim. As per Allen's (2017b) dissertation, two techniques were used to reconstruct precipitation from foliar assemblages: a univariate approach, leaf area analysis (LAA; Wilf et al., 1998) and a multivariate approach, Climate Leaf Analysis Multivariate Program (CLAMP; Wolfe 1993). LAA is based on the correlation between mean leaf area and annual precipitation, related to transpiration. Leaves with higher surface area to volume ratios transpire more during gas exchange; these larger leaves are typically found in wet areas (Wilf et al., 1998). Leaves in drier climates have smaller leaf area to volume ratios, as they do not have as much plant available water accessible to transpire (Wilf et al., 1998). CLAMP uses 31 morphological characters on at least 20 species of woody "dicots" from any given site to reconstruct eleven aspects of climate, including mean annual precipitation, as well as mean annual temperature (MAT, comparable to mean annual air temperature – MAAT – discussed in this study; Wolfe 1993; Spicer et al., 2009). This method is premised on the relationship between these morphological characters in modern flora and corresponding climate parameters.

Physiognomic techniques including CLAMP and leaf margin analysis (LMA; Wilf 1997) were used to calculate mean annual air temperature. LMA uses the correlation between MAAT and the proportion of untoothed to total (untoothed + toothed) species in a local flora (Wolfe 1979; Wilf 1997; Wing & Greenwood 1993; Peppe et al., 2011). See Allen (2017b) for additional reconstruction details based on floral assemblages.

### 3.3.4 Paleoclimate and Paleoenvironmental Reconstructions using Organic and Inorganic Geochemistry

Mean annual precipitation was reconstructed using Chemical Index of Alteration minus potassium (CIA-K; Equation 5; 6; Sheldon et al., 2002; error $\pm$ 182 mm yr$^{-1}$), modified from CIA to control for potassium metasomatism in paleosols (Maynard 1992; Ennis et al., 2000; Sheldon et al., 2002). Mean annual air temperature was calculated using PWI (Equation 7; error of $\pm$ 2.1 °C; Gallagher & Sheldon 2013). We applied the empirical relationship between $\delta^{13}C_{plant}$ and $\delta^{13}C_{atm}$ values found by Arens et al. (2000; Equation 8; $R^2 = 0.34$, $p < 0.001$) and used $\delta^{13}C_{plant}$ values of all individual fossils to reconstruct generalized, non-taxon-specific $\delta^{13}C_{atm}$ values. We compared this reconstructed value based on a generalized equation with

reconstructed values based on species-specific carbon isotope discrimination values (as measured in Cornwell et al., 2018), using fossil *Lygodium* and *Populus* to reconstruct $\delta^{13}C_{atm}$ values based on taxon-specific parameters (e.g., Stein et al., 2019; Stein et al., 2021).

$$CIA - K = \frac{Al_2O_3}{Al_2O_3 + Na_2O + CaO} * 100 \tag{5}$$

$$MAP = 221e^{0.0197(CIA-K)} \tag{CIA-K for paleosols; 6}$$

$$T\ (°C)_{PWI} = -2.74 * \ln(PWI) + 21.39 \tag{7}$$

$$\delta^{13}C_{leaf} = 1.10\ (\delta^{13}C_{atm}) - 18.67 \tag{8}$$

Holdridge life zones are ecoregions classified by water availability and temperature, that can be further subdivided into successional stages reflecting land use, disturbance history, latitude, altitude (Holdridge 1967; Lugo et al., 1999). The parameters for each life zone are calculated based on potential evapotranspiration and humidity provinces (Holdridge 1967; see Appendix D). Similar metrics that use evapotranspiration and precipitation to quantify ecosystems into "floral humidity

provinces" based on paleosol measurements, have been established more recently by Gulbranson et al. (2011; see Appendix D). See supplemental materials for methodology used to determine Holdridge life zones and Floral Humidity Provinces for paleosols (this study) and previously published floras (Leopold & MacGinitie 1972; Roehler 1993; Wing et al., 2005; Smith et al., 2008; Wing & Currano 2013; Allen 2017a/b).

**4 Results**

**4.1 Geochronology**

Single crystal sanidine $^{40}$Ar/$^{39}$Ar analyses of four sampled beds yielded ages for the middle and top of the Blue Rim escarpment that are broadly consistent with deposition during the early Eocene (Fig. 1; Table 1; Table S2). Two samples (BR-3 and BR-4) of a horizon containing pumice clasts and biotite grains taken from the base and middle of the 'blue-green

marker' bed yielded similar coherent young populations of Eocene apparent ages (Figs. 4a; 8a), which are mixed with a subsidiary population of older, presumably detrital or xenocrystic grains (Fig. 8b). Sample BR-3 was collected at the base of

the main blue-green marker layer, just above the UF 15761S plant quarry (elevation 2053 m; Allen 2017b), whereas sample BR-4 was collected from the lower part of the blue-green layer in the 2014/UF 19297 stratigraphic section at 2056 m (Allen 2017b). Twenty seven of forty fusions of sanidine from these beds form a coherent population that yields a weighted mean age of 49.29 ± 0.18 Ma (MSWD = 1.08; Table 1), which we interpret to reflect the best estimate of the age of deposition. This age is consistent with the Blue Rim being coeval with Bridger A/1b (Fig. 1).

Single fusions of sanidine from two sand beds (samples BR-5 and BR-6) collected near the top of the escarpment (elevation 2091 m; Allen 2017b) yielded a greater proportion (31 of 38) of older detrital and/or xenocrystic ages than occur in samples of the blue-green marker. Nevertheless, the seven youngest grains from the uppermost sample BR-6 form a coherent population that yields a weighted mean age of 48.48 ± 0.60 Ma (MSWD = 0.60), which can be interpreted to be a maximum depositional age. This imprecise age suggests that the uppermost Blue Rim could be as old as Bridger B/B-2, or as young as Bridger D/Br-3 (Fig. 1). Altogether, new geochronology indicates that the stratigraphy between the blue-green marker bed and sand beds likely spans Bridger B, with the uppermost part of the Blue Rim Escarpment being time equivalent to Bridger C or D/3 (Fig4, 5, 6). This constrains the age of the lower plant horizon to be slightly older than 49.29 Ma (Allen 2017b), whereas the upper plant horizon is likely equivalent to Bridger Br-2. Detrital or xenocrystic grains not included in weighted means discussed above yield a combination of Phanerozoic and Proterozoic apparent ages which are consistent with detrital feldspar grains sampled from volcaniclastic strata of the Sand Butte bed of the Laney Member (Green River Formation, Smith et al., 2008; Chetel et al., 2011). The Sand Butte bed forms thick (> 40 m) deltaic foresets that prograded into and filled Lake Gosiute and parts of Lake Uinta from northwest to southeast (Surdam & Stanley, 1980; Smith et al., 2008; Chetel & Carroll 2010). These deposits have been hypothesized to represent the inland delta or megafan of the Idaho paleoriver which drained areas as far west as central Idaho (Chetel et al., 2011).

## 4.2 Paleosol descriptions

Six profiles were sampled laterally from a single paleosol at the base of the Blue Rim escarpment (located at 1 m in the stratigraphic column; Figs. 4—7). Paleosol profiles (Fig. 7, S1) typically consisted of a silty and/or sandy brown, yellow A-horizon over a parent material C-horizon of green-grey silty mudstone from ~20 to 112 cm below the surface. Paleosol #1 was

missing a B-horizon due to erosion, while paleosol #4 was missing an A horizon, likely truncated during burial. Typically, each profile was lighter colored in upper horizons and darker in lower B- and C-horizons. In the paleosol profiles sampled, every A-horizon but one, and several upper B-horizons, had root traces, kerogenized roots, and/or rhizoliths. We observed drab-haloed roots in paleosols #1 and 4. Paleosol #2 had vertical burrows of up to 1 cm diameter and ~3 cm length, and paleosol #1 had visible peds (Table S1; Fig. 3a). These soils were well-developed Inceptisols based on features, textures, and extrapolation from the local flora (Fig. 2a; Fig. S1; S5; Table S1).

## 4.3 Paleosol geochemistry

On average, the lateral extent of the paleosol found at the base of the Blue Rim stratigraphic section had A-horizon CIA-K values in the 50s with a maximum of ~60 in B- and C- horizons (consistent with expectations for CIA-K values based on past work; Sheldon et al., 2002; Sheldon & Tabor 2009; Fig. S1b). Ti/Al ratios were constant throughout, ranging from 0.040 to 0.045, typical of values of mudstones and sandstone parented materials (Sheldon & Tabor 2009), like those seen in sediments throughout the Blue Rim escarpment (Fig. S1c; Table S2).

Tau (used to measure mobile element transport) was calculated for soil profiles following Chadwick et al., (1990; Equations 2 and 3) and displayed in Supplemental Fig. S2(a-f). Overall, tau values for K, Mg and Na all ranged from 0.0 to -0.5, and tau values for Ca ranged from 0.0 to -1.0, except in paleosol #2 (which was extremely high in Ca), as is typical for Inceptisols. Tau values for Rb and Fe were generally also between 0.0 and -0.5, though this was less consistent between profiles. To note, paleosol #1 (19BRWY1; Fig. 7) was excluded for paleoclimate reconstructions due to the lack of presence of the B- horizon (we identified this soil as an Entisol, which cannot be used for climate reconstructions; see Fig. 7 for location). Paleosol #2 (19BRWY2) was also excluded for climate reconstructions, due to the high % Ca, likely of carbonate origin as this site was reactive to HCl. Likewise, the CIA-K values for 19BRWY1 and 19BRWY2 B- horizon specimens were not reasonably different than the parent material, indicating they were not in equilibrium with the environment (e.g., Sheldon & Tabor 2009). Based on both field taxonomy and these geochemical results (see Supplemental Table S3), paleosols #3-#6 are identified as Inceptisols (Soil Survey Staff, 2014).

### 4.4 Sedimentary geochemistry

All Blue Rim escarpment geochemical data has been published at Mendeley Data Repository doi: 10.17632/z6twpstz4r.3. With a few individual outlier analyses, proxies for leaching intensity (Ba/Sr, and sum bases/Ti; Fig. 4b,c), hydrolysis (sum bases/Al; Fig. 4d), and measurements of weathering (CIA; Fig. 4e) are consistent throughout the section. Proxies for provenance also remained nearly constant (Fig. 5a,b) as did proxies for parent material (Fig. 5c,d). Organic carbon weight % was high in the same locations throughout the section as CIA, and % N was low throughout the section. $\delta^{13}C_{org}$ values were most depleted in the sections with highest % C and N.

### 4.5 Flora

The identifiable fossils from the 2019 field excursion sampled specifically for organic isotope analyses included multiple compression fossils of *Lygodium kaulfussi* (climbing fern, family Lygodiaceae; Manchester & Zavada 1987), as well as one specimen of *Asplenium* sp. (fern, family Aspleniaceae; as described in Allen 2017b), an example of cf. *Populus cinnamomoides* (poplar, family Salicaceae; Manchester et al., 2006), one specimen assigned to cf. *Cedrela*, (undefined species; mahogany, family Meliaceae; Fig. S5), several dense leaf mats, and assorted twig and branchlet fossils were recovered. These specimens were collected from the same strata as the lower horizon (e.g., UF 15761N, Allen 2017b), located 26 m on the stratigraphic column included in this study (Figs. 3–5). There were also fragments of several unidentified monocots preserved, though no isotope analyses were run on these fossils.

### 4.6 Climate

Mean annual precipitation (MAP) values reconstructed using CIA without potash (CIA-K) on paleosol B-horizons (Equation 6) ranged from 608–1167 mm yr$^{-1}$, with an average of 845 mm yr$^{-1}$ (±181 mm yr$^{-1}$; ($n = 6$) (Equation 6; Fig. 6; S6). The lowest estimated MAP value (288 mm yr$^{-1}$) was excluded due to high % Ca (10.25%) in the B horizon of paleosol 2, skewing CIA-K calculations by artificially minimizing the ratio of Al to other metals in the calculation (see Dzombak et al., in review for discussion of this issue). Mean annual air temperature values (MAAT) reconstructed using PWI on paleosol B-horizons (Equation 7; standard error of (± 2.1 °C)) ranged from 10.4 to 12.0 °C (± 0.7 °C standard deviation of all values

from the same paleosol, or temperature reproducibility), with an average of 11.0 °C ($n = 6$ profiles). Temperature reproducibility from these paleosols falls within the standard error of the paleothermometer model.

A wide range of $\delta^{13}C_{atm}$ values were reconstructed from $\delta^{13}C_{leaf}$ from the 34 individual leaf fossils (2019 collection) using a generalized relationship (Arens et al., 2000). Reconstructions using the generalized Arens et al., (2000) model were done on all 34 individual fossil leaves, even though many of these were unidentified. Additional species-specific tests were done on all samples of *Lygodium*, cf. *Cedrela* and *Populus* fossils using isotope discrimination values from extant plants of these genera. Thirty eight percent ($n = 13$) of the 34 $\delta^{13}C_{atm}$ values reconstructed using the generalized Arens et al., (2000) model suggested a $\delta^{13}C_{atm}$ value of between −5.32 and −5.82‰. Fifty six percent of these reconstructed values were between −5.0 and −6.0‰ ($n = 19$; Equation 8; Fig. 9). One limitation on this reconstruction is that it does not account for species-specific isotope discrimination behaviour that varies taxonomically (Beerling & Royer 2002; Stein et al., 2019; Sheldon et al., 2020). Values on the fringes of these values (44%) may be so extreme after having experienced diagenesis of certain compounds while others were left behind, skewing the isotopic values to be representative of the compounds and not the bulk tissue (Beerling & Royer 2002; Tu et al., 2004). Using identified *Lygodium,* cf. *Cedrela* and *Populus* fossils ($n = 8$ total), we applied the taxon-specific isotope discrimination principle and reconstructed an average value of −4.40‰ (minimum value of −5.23‰ and maximum value of −3.83‰). These reconstructions were based on isotope discrimination values of 19.99‰ for *Lygodium* and 20.05‰ for *Populus* (as reported in modern isotope analyses by Cornwell et al., 2018).

## 5 Discussion

### 5.1 Geochronology

The new $^{40}Ar/^{39}Ar$ dates presented here for the blue-green marker bed and sand beds above the floral quarries suggest that the Blue Rim section likely spans ca. 49.5 to 48.5 Ma, making it slightly younger than the EECO. Assuming constant sedimentation rates based on a linear interpolation between ages of 1 m per ~23 ka (or ~44 μm yr$^{-1}$, slightly slower than accumulation rates of 65 μm yr$^{-1}$ +19/-12 in the Laney Member below (e.g., Smith et al., 2010), the newly described paleosols are roughly 684,300 years older than the blue-green marker bed, or 49.97 Ma. The Wilkins Peak Member, and part of the Laney Member of the Green River Formation underlie the Blue Rim strata; thus, the approximate age for the paleosols based

on constant sedimentation rate is consistent with Blue Rim being younger than ~50 Ma, when the Wilkins Peak Member transitioned to the Laney Member (Smith et al., 2015; see Fig. 1).

**5.2 Stratigraphy, provenance, and weathering**

Detrital feldspar geochronology and geochemistry strongly suggest that provenance remained constant throughout accumulation of the stratigraphic section, which can be interpreted to mean that the material did not systemically change across deposition, and geochemical proxies used to reconstruct climate are not affected by provenance shifts. A sudden 6 ‰ decrease in $\delta^{18}O$ was previously observed in micritic lacustrine carbonates within sections of the Green River Formation, and

410 it was hypothesized that this could be due to a sudden change in river capture to include a river with isotopically depleted waters from different headwaters (e.g., Norris et al. 1996; Norris et al., 2000; Doebbert et al., 2010). With new river catchments, there is a potential change in sedimentological inputs that could overwrite climate signals, so these geochemical proxies provide assurance that the climate interpretations based on geochemical proxies are not actually changes in allochthonous materials. Assumptions about provenance were supported by parent material data, which showed that all

415 sources were primarily sedimentary. The consistent Ti/Al, U/Th, and La/Ce ratios correspond to constant parent material throughout the one million years section covered by the Blue Rim stratigraphic column, demonstrating that basin-scale hydrology was likely not reorganized during this time. The one exception to constant parent material and provenance ratios is the anomalously high U/Th ratio in the blue-green marker bed (0.87). This proxy is redox-sensitive, so this anomalously high U/Th ratio is due to the preferential redistribution and accumulation of U in this section, as Th is insoluble and

420 immobile (Pett-Ridge et al., 2007; Sheldon & Tabor 2009), intuitive with the Blue-green marker color. Weathering and leaching were highest in the sections where there was high carbon content; this correlation could be due to organic acids produced by plants in the ecosystem (as represented by % C present) that contribute to chemical weathering, or differences in taphonomic histories of samples. More compound-specific analyses of organic compounds would be needed to determine if and how plant influence is contributing to weathering in high % C sections (Fig. S4; $R^2 = 0.20$; p-value = 0.01; Ong et al.,

1970; Berner 1992).

### 5.3 Global and regional climate

Generally, the early Eocene of North America had widespread wet forests comparable to modern temperate and subtropical forests due to global warmer, wetter conditions (Leopold & MacGinitie 1972; Wing & Greenwood 1993; Greenwood & Wing 1995; Inglis et al., 2017; Murphey et al., 2017). Depending on latitude of site, other studies indicate slightly to moderately warmer conditions temperature reconstructions from Blue Rim (Allen 2017a/b). Generally, mean annual air temperatures reconstructed from other mid to high- latitude sites between 36 and 80 °N have ranges from 35°C (36 °N) to 8°C (80 °N; using leaf margin analyses and oxygen isotopes; Fricke & Wing 2004).

More locally, temperature reconstructions of other Wyoming formations based on floral assemblages from earlier in the Eocene were higher than paleosol-based temperature reconstructions at Blue Rim (i.e., paleosol-based estimates were 12 °C, while other temperature reconstructions were higher; Wilf 2000, Wing et al., 2005; Hyland et al., 2018; Fig. 11), though contemporaneous plant-based reconstructions were more comparable ~12 °C (Leopold & MacGinitie 1972; Allen 2017b; Fig. 11). Summer months were warm, and winter months mild, with temperatures from 18 to 34 °C and 4 to 7 °C respectively (based on $\Delta_{47}$; Kelson et al., 2017; Hyland et al., 2018).

The location of Blue Rim appears to have been a "wet forest" at the time of deposition, as calculated from temperature and precipitation estimates from paleosols and using leaf physiognomic techniques (see supplemental methods for details; Fig. 10). The paleosol-based temperature and precipitation results are within error of flora-only estimates based on contemporaneous Blue Rim reconstructions by Allen (2017b) and Wilf et al., (2000; other southwest Wyoming localities including Little Mountain, Niland Tongue, Sourdough, Latham, Wasatch Main Body, Big Multi Quarry, and Bison Basin) and the ecosystem deposited is similar to Wing et al.'s (2005) ecosystem reconstruction of the Paleocene-Eocene thermal maximum (PETM; Bighorn Basin in north-central Wyoming; McInerney & Wing 2011; Fig. 11; Fig. S7). Indeed, fossils collected for organic analysis at Blue Rim escarpment included mineralized trunks and fern and angiosperm leaf macrofossils including: *Lygodium kaulfussi*i, *Asplenium*, *Populus cinnamomoides,* cf. *Cedrela,* as well as dense leaf mats. These collected fossils (sampled at 26 m on the stratigraphic section, housed at ESS laboratory at the University of Michigan) are taxonomically comparable to fossils found by Allen (2017b; Bridger Formation) and MacGinitie (1969; Green

River Formation). The flora sampled are characteristic of a mesic, forested environment (e.g., wet forest; Hamzeh & Dayanandan 2004; Hamzeh et al., 2006).

While the leaf physiognomic and paleosol-based estimates are modestly different, we cannot differentiate between whether there was a slight increase in both MAAT and MAP or whether climate was generally steady throughout the reconstructed portions of the sections, because calculated values were within error. One possible explanation for the slight discrepancy between leaf physiognomic and soil-based temperature reconstructions at Blue Rim could be related to the soil taxonomy; the PWI tool used to reconstruct temperature was calibrated for Inceptisols, Alfisols and Ultisols. However, none of the Inceptisols sampled to calibrate this proxy were from mean annual air temperatures >12 °C (Gallagher & Sheldon 2013). Therefore, it is possible that temperature reconstructions based on PWI for these Inceptisols are underestimates. Due to the complexities in the formation of soil related to seasonality of precipitation and temperature, seasonal bias has been found in paleosol-reconstructions based on carbonates (e.g., Kelson et al., 2020) but B-horizon bulk geochemical data is in equilibrium with the environment and takes so long to form, therefore is not resolved enough to be affected by seasonality (e.g., Sheldon et al., 2002). Seasonal bias that would not be seen in leaf physiognomic techniques, either.

Leaf physiognomic proxies could contribute to the discrepancy as well; CLAMP has been cited as often producing overestimates of precipitation (Wilf et al., 1998; Allen 2017b). This is exacerbated when there are fewer than 25 morphotypes available, which was the case at Blue Rim (Wolfe 1993; Spicer et al., 2009; Allen 2017b). CLAMP estimates may be less accurate due to the threshold number of morphotypes used at Blue Rim escarpment (20); the number of morphotypes used for CLAMP temperature reconstructions was exactly the minimum recommended value (Allen 2017b).

Regardless of the cause of discrepancy or if it represents modest actual change versus stasis, this study demonstrates the importance of the holistic approach that combines both types of proxies. However, based upon their close statistical agreement (Figure 6) and implications for the overall ecosystem (Figure 10), we interpret the overall climate as relatively steady (± 5 °C) on 100,000 year or more timescales during this interval.

**5.4 Maintaining Warmth in the Paleogene**

The maintenance of extended warmth and its relation to elevated $CO_2$ during the early Eocene is debated (Hyland & Sheldon 2013; Anagnostou et al., 2016; Gutjahr et al., 2017; Cramwinckel et al., 2018), which emphasizes the importance of $\delta^{13}C_{atm}$ values that can help to contextualize potential $CO_2$ sources. Some scientists invoke the destabilization of deep-sea methane hydrates as the mechanism for $CO_2$ increase (e.g., Dickens 2011), while others pinpoint volcanic emissions (Reagan et al., 2013; Gutjahr et al., 2017; Jones et al., 2019) and reduced silica weathering (Zachos et al., 2008; Lunt et al., 2011).

These mechanisms occur on very different timescales (e.g., methane has a lifetime of 12 years in the atmosphere; Schiermeier 2020), and thus require vastly different environmental processes and landscapes. It is possible to determine changes in the source of atmospheric $CO_2$ by examining the isotopic signature of the atmosphere; volcanic $CO_2$ has an isotopic composition of −5.4 ‰ (Deines 1992) while methane hydrates are far more depleted (e.g., −64.5 to −67.5 ‰ off the coast of central Oregon; Kastner et al., 1998). As mentioned above, based on the comparative Arens et al., (2000) model,

$\delta^{13}C_{leaf}$ values can be used to infer $\delta^{13}C_{atm}$ values. These values are also important parameters in models that reconstruct additional environmental variables, such as the concentration of atmospheric $CO_2$ using paleosol carbonates (Cerling et al., 1991; Cerling 1992) or using stomatal parameters (Franks et al.. 2014). Our large sample size of distinguishable leaf fossils from a single horizon ($n = 34$) allowed us to statistically determine the most likely $\delta^{13}C_{atm}$ value based on the general relationship proposed by Arens et al., (2000) to be between −5.32 and −5.82 ‰. Analyses with taxon-specific

reconstructions ($n = 9$ total specimens of these genera: *Lygodium,* cf. *Cedrela* and *Populus* fossils), using extant members of these genera to determine isotope discrimination and reconstruct the atmosphere (as published in Cornwell et al., 2018) had an average value of −4.82 ‰ (±0.92 ‰ standard deviation). Both generalized and taxon-specific reconstructions were within error of the isotopic value of mantle $CO_2$ (−5.4 ‰; Deines 1992), and comparable to the value reconstructed for ~49 Ma using benthic and planktonic foraminifera, −5.4 ‰ (Tipple et al., 2010). Although we cannot rule out short term

perturbations of other C pools (e.g., methane hydrates during the PETM, Foster et al., 2018; shift in carbon from the deep ocean due to changes in the carbon compensation depth; Zeebe et al., 2009; Pälike et al., 2012; Zeebe 2012), these results support previous findings indicating a long-term volcanic source of $CO_2$ that drove long-term warmth in the Paleogene. However, this observation is incomplete without full constraints regarding the calcite compensation depth and the distribution of carbon isotopes in carbon ocean chemistry (e.g., Komar et al., 2013), both of which are not constrained in the

scope of this study. As per other studies, the early Cenozoic period of elevated rates of volcanism (locally in the Rocky

Mountains; e.g., Challis volcanics; Fig. 3a; Chetel et al., 2011; as well as globally with the North American Igneous

Province; ~61 to ~50 Ma; e.g., Meyer et al., 2007; Storey et al., 2007) can account for this.

## 6 Conclusions

The age findings contained in this study constrain the time for the Blue Rim wet forests to be slightly younger than

previous estimates, with the upper half of the section clearly deposited after the EECO. Based on dating and sedimentation

rates, the lower part of the section could overlap with the end of the EECO, but there is an absence of observable changes in

proxy records. During the EECO, the Blue Rim escarpment received between 608–1167 mm yr$^{-1}$ of precipitation, likely

related to different moisture regimes, and was a productive paratropical (non-equatorial tropical) forest. Although

reconstructed temperature and precipitation values using paleosol and sedimentary geochemistry are lower than published

values reconstructed from flora, the values from all the proxies all fall within error of one another (see supplemental

materials). Furthermore, the Holdridge life zones and floral humidity provinces calculated for both leaf physiognomic-based

and geochemistry-based reconstructions are comparable, pinpointing this region as a warm, wet forest at ~49 Ma.

The new age constraints, evidence from leaf fossils, and inorganic and organic geochemical proxies at Blue Rim

escarpment make it possible to reconstruct the depositional environment of the central region of Lake Gosiute in an

unprecedented way. The consistent Ti/Al, U/Th, and La/Ce ratios to determine provenance and parent material throughout

the Blue Rim stratigraphic column demonstrate that the hydrological and sedimentological inputs remained stable for this

location throughout the ~one million years spanned by this section. The use of multiple proxies to cross-compare sites is

under-utilized in paleoclimate reconstructions but allows for an improved understanding of regional and more broad-scale

climate regimes. Based on ample floral and geochemical data, ~49.5–48.5 million years ago – after the peak of the EECO –

southwest Wyoming was a warm, wet forest atop deltaic deposition (Fig. 3a; Chetel et al., 2011) with little to no frost and

mild temperatures, in agreement with previous work based only on fossil floras (e.g., MacGinitie 1969; Wilf 2000) or

paleosols (Hyland et al., 2018) individually.

## 7 Author contributions

NDS and RAS conceived and designed the study. RMD and RAS collected the data in the field. MES, SEA and BRJ contributed data. NDS and MES were responsible for funding acquisition, and RAS acquired funding for field work. NDS was responsible for project administration, provision of resources and supervision to RAS. RAS was primarily responsible for the investigation and performed many of the visualizations. RMD, MES, and SEA contributed visualizations as well. RAS and NDS were responsible for the original draft, and all authors (RMD, RAS, MES, SEA, BRJ) were responsible for review and editing.

## 8 Acknowledgments

We thank Nikolas C. Midttun for assistance measuring, creating, and sampling the stratigraphic column at the Blue Rim escarpment in June 2019. We thank Steven R. Manchester for personal communications regarding the location of floral fossil quarries at Blue Rim. We thank Selena Y. Smith for personal communications and consultation regarding fossils and for access to her camera and camera stand for fossil photographs. We acknowledge Naomi E. Levin, Christopher J. Poulsen and Gretchen Keppel-Aleks for their feedback on this manuscript. This work was partially funded by NSF Award #1812949 to Nathan Sheldon and Michael Smith, and an Evolving Earth Graduate Research Grant to RAS.

## 9 Code/Data Availability

The geochemical dataset for this manuscript can be found at Mendeley Data Repository under the title "Blue Rim escarpment geochemical data", Mendeley Data, V1, doi: 10.17632/z6twpstz4r.1.

## 10 Competing Interests

The authors declare no competing interests.

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

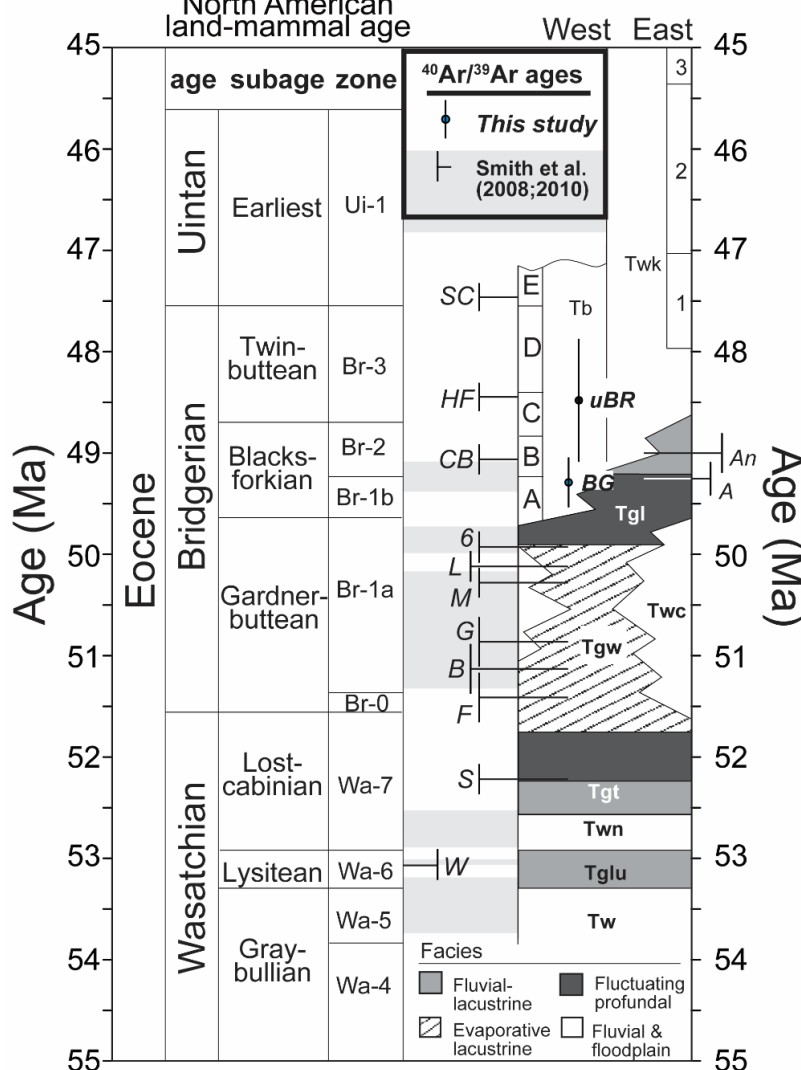

**Figure 1: Age model of the Eocene Wyoming. North American Land Mammal Age (NALMA) is shown for context.** Upper Blue Rim is shown as "uBR" and the blue-green marker is depicted as "BG," and both are shown with dots and error bars. Dates from Smith et al. (2008; 2010) are shown in perpendicular lines. The abbreviations included are designated as follows. Green River Formation: *Tglu* Luman Member, *Tgt* Tipton Member, *Tgl* Laney Member. Wasatch Formation: *Tw* "Main body," *Twn* Niland Tongue, *Twc* Cathedral Bluffs Member, *Tb* Bridger Formation, *Twk* Washakie Formation. Tuff beds as measured in Smith et al. (2008; 2010) are denoted as: *W* Willwood, *S* Scheggs, *R* Rife, *F,* Firehole, *B* Boar, *G* Grey, *M* Main, *L* Layered, *6* Sixth, *A* Analcite, *CB* Church Butte, *HF* Henrys Fork, and *SC* Sage Creek.


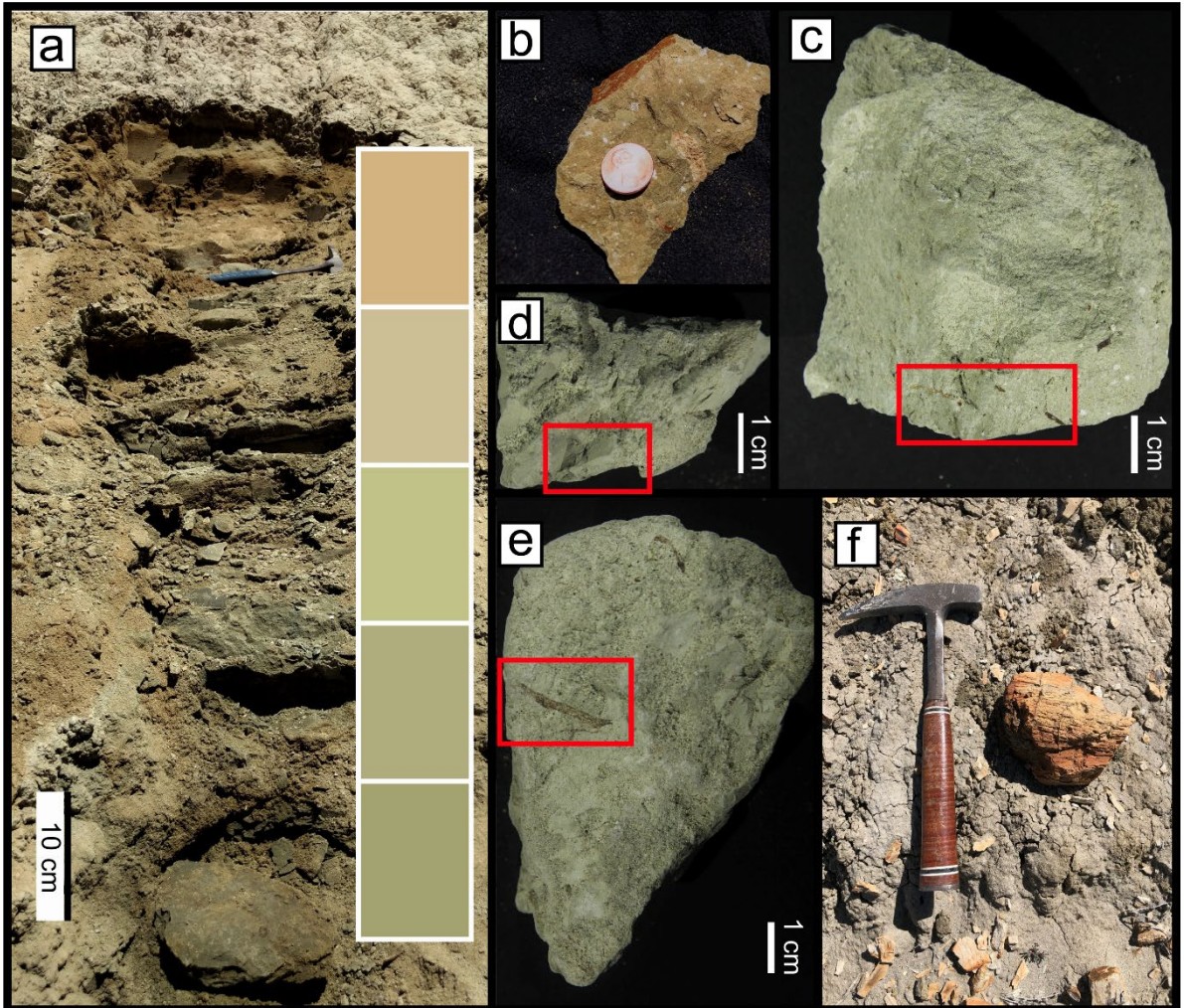

**Figure 2: Paleosol images and features.** (a) Full profile of 19BRWY3 with rock hammer for scale, (b) shows drab-haloed root trace from 19BRWY-2UB, (c) A-horizon fine organic rootlets in red box from 19BRWY2UA, (d) slickensides on 19BRWY1UA, (e) rhizoliths from 19BRWY1UA, and (f) mineralized wood with rock hammer for scale, excavated ~3 m
above 19BRWY1.

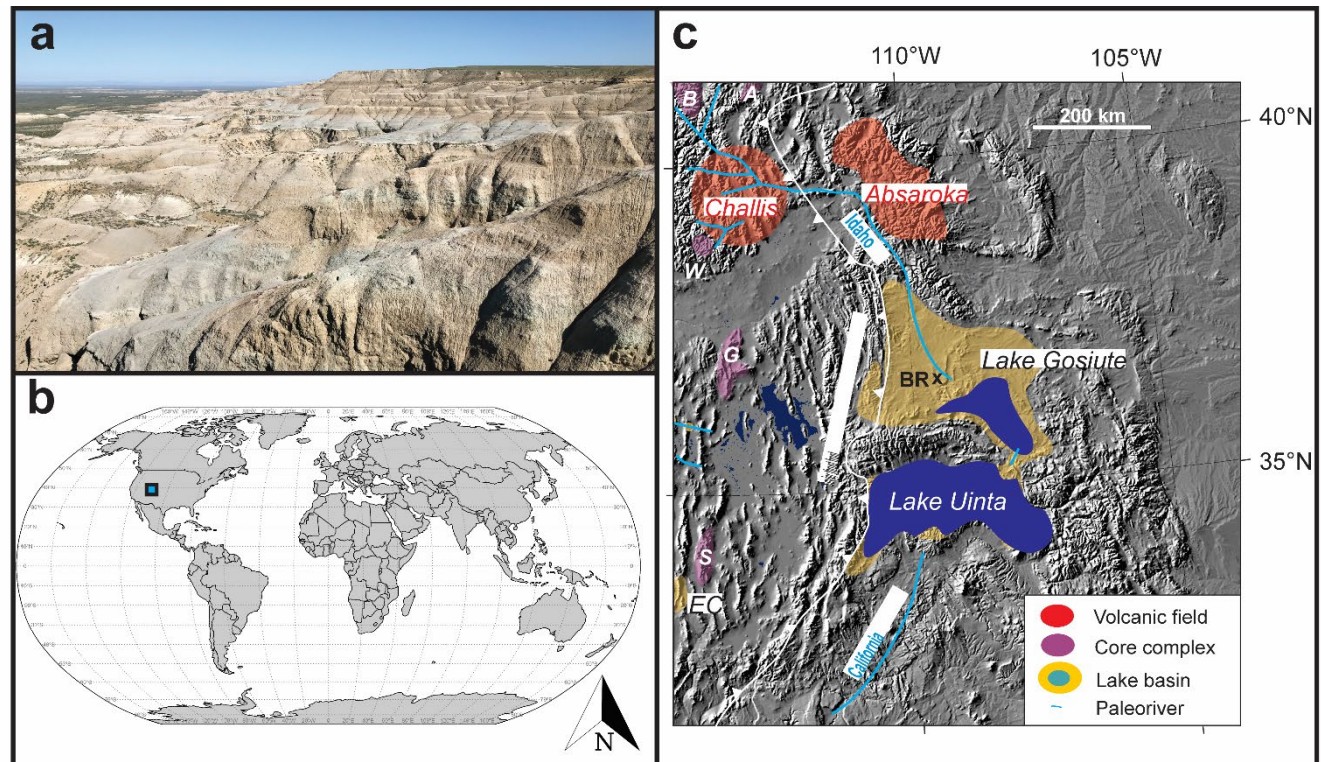

**Figure 3: Map and profile of Blue Rim escarpment.** (a) Landscape image of the escarpment from the uppermost strata. (b) Location of Blue Rim escarpment (blue square) at present, in context of the present tectonic configuration of the world using a Robinson projection map. (c) Shaded relief map of the North American Cordillera showing the paleogeographic position of the Blue Rim (BR) relative to major paleorivers, lake basins, and tectonic elements. EC refers to early Bridgerian Elderberry Canyon local fauna of Emry (1990). Core complexes occur near the Cordilleran paleodivide: *B* – Bitterroot; *A* – Anaconda; *W* 950 – Wildhorse; *G* – Albion-Raft River-Grouse Creek; *S* – Snake. Note that the Bridger Formation at Blue Rim represents the topsets of the Sand Butte delta (cf. Smith et al., 2008). Paleorivers summarized from Henry et al. (2012), Dickinson et al. (2012) and Chetel et al. (2011).

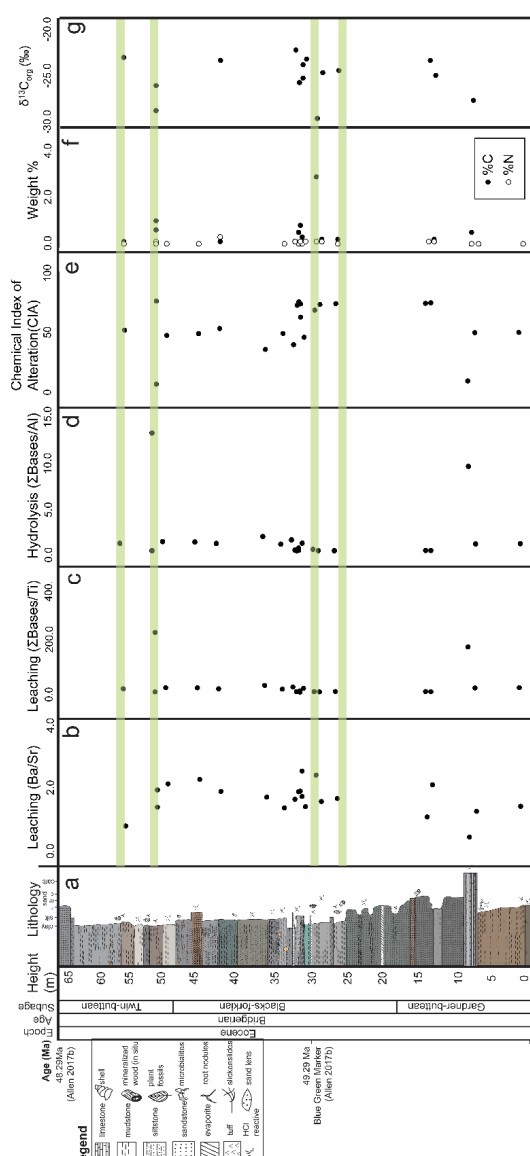

**Figure 4: Stratigraphy with sedimentary geochemistry.** (a) Stratigraphic column at the Blue Rim escarpment of the Bridger Formation. Geochemistry including (b) leaching, calculated using molar ratios of Ba/Sr, (c) leaching, calculated using the ratio of the sum of bases to Titanium, (d) hydrolysis, calculated using the ratio of the sum of bases to Aluminum, (e) chemical index of alteration to measure weathering (Equation 1), (f) weight % Carbon (black circles) and % Nitrogen (white circles), (g) $\delta^{13}C_{org}$ values. Light green transparent areas show stratigraphic levels containing plant fossils.

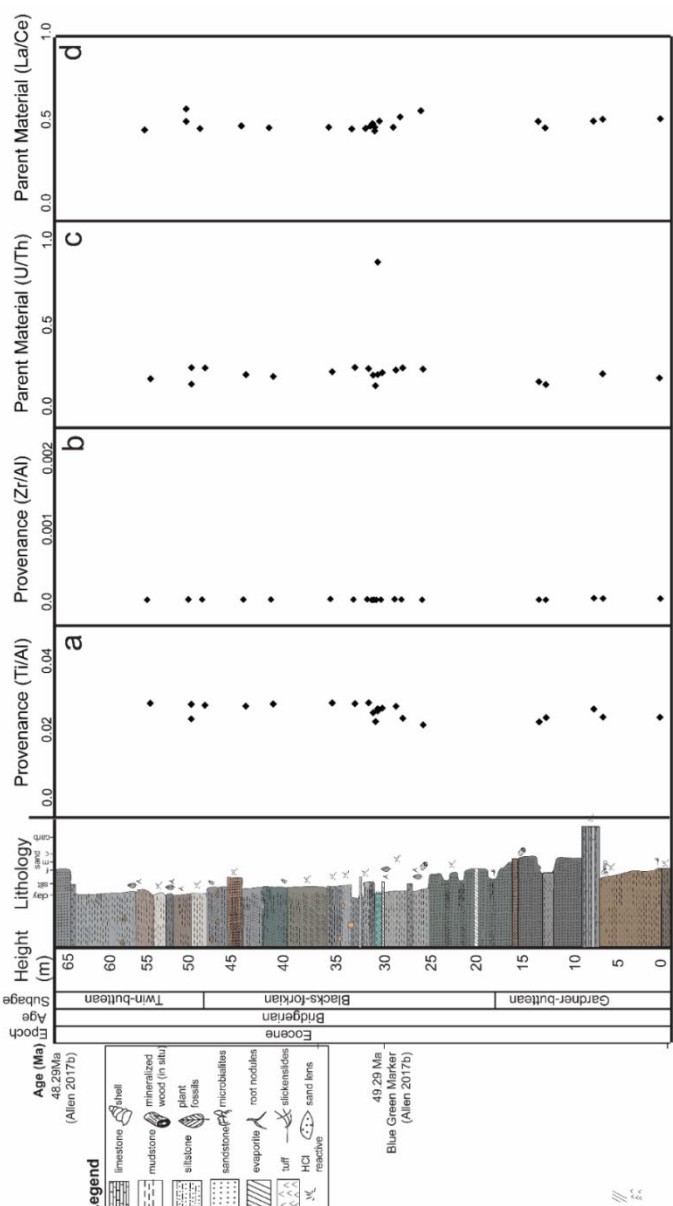

**Figure 5**: **Stratigraphy with parent material and provenance proxies.** (a) Provenance using molar ratios of Ti/Al, (b) provenance using molar ratios of Zr/Al, (c) parent material using molar ratios of U/Th, (d) parent material using molar ratios of La/Ce.

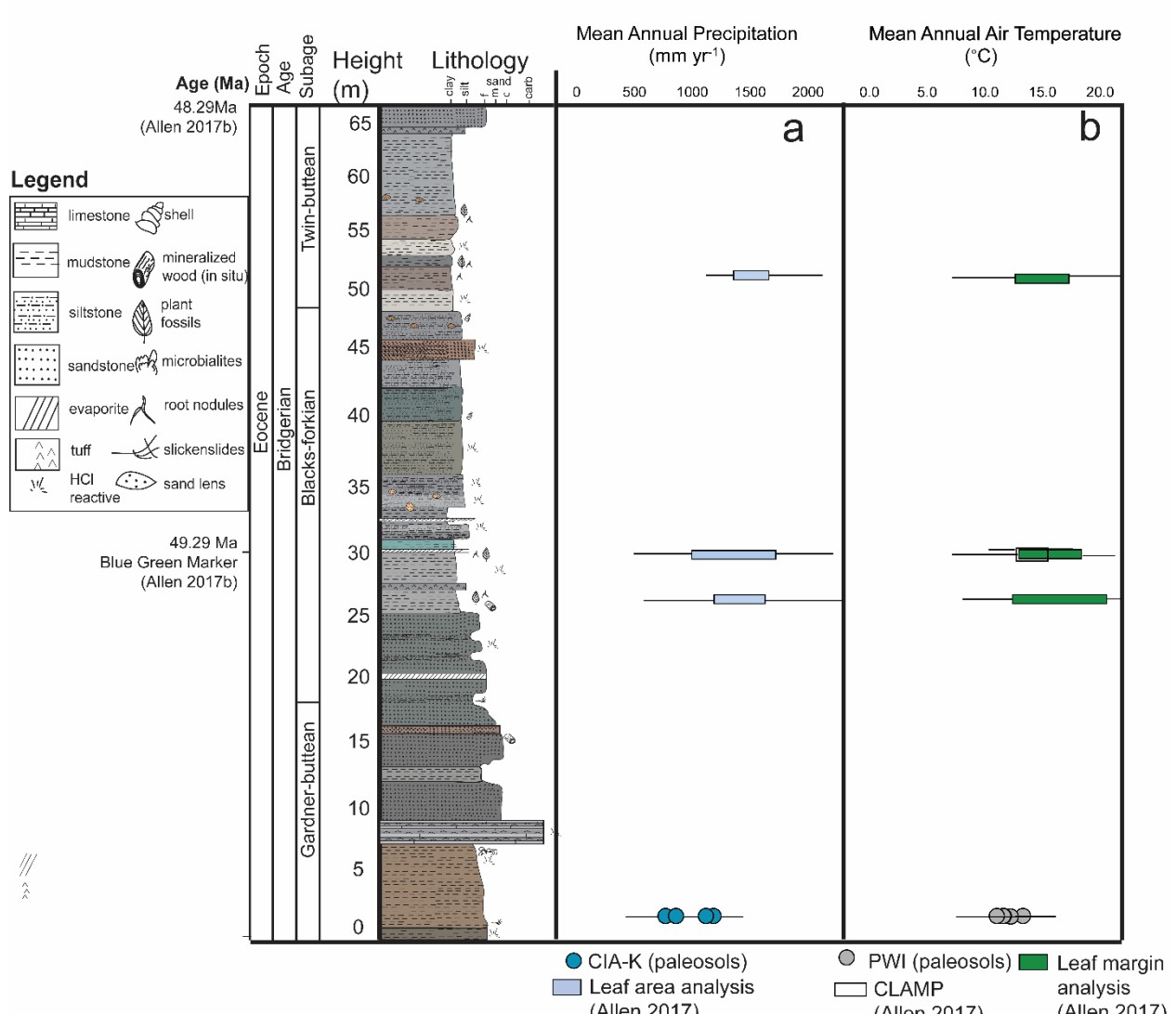

**Figure 6: Stratigraphic column with climate proxies.** (a) Reconstructed mean annual precipitation (mm yr$^{-1}$), and (b) mean annual air temperature (°C). The boxes are ranges reconstructed for physiognomic proxies, and the circles are values reconstructed using paleosols. The lines for both are established errors for the proxies used, based on the initial proxy calibrations. Errors were calculated for each estimate, and the error bars span the total range of calculated error. More information on error can be found within these proxy calibrations (e.g., Wilf 2000; Sheldon et al., 2002; Spicer et al., 2009; Gallagher & Sheldon 2013).


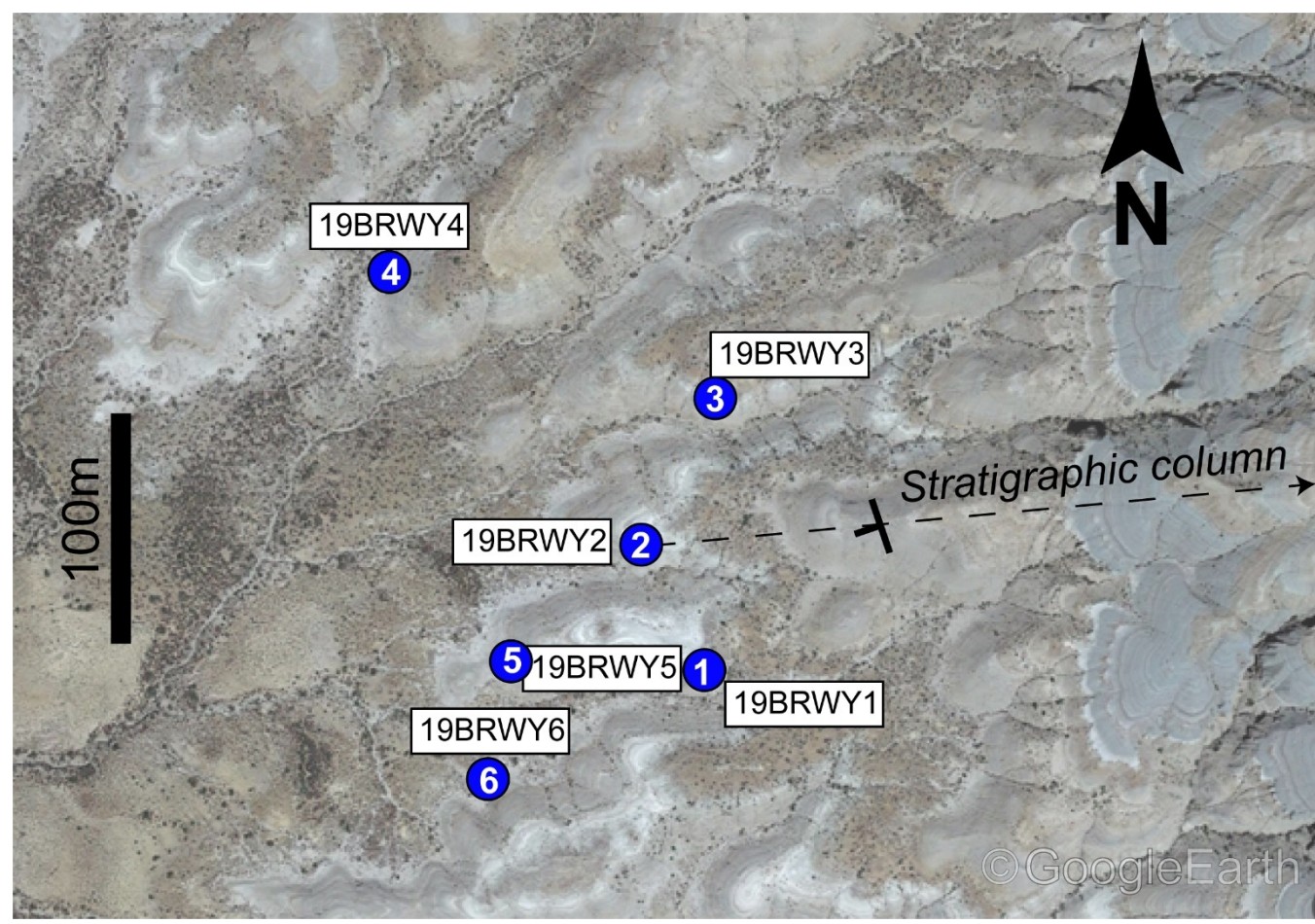

**Figure 7: Lateral extent of paleosols.** Included are paleosols 19BRWY1-6, atop image of Blue Rim escarpment topography, with a 100 m scalebar. Paleosol numbers are noted in white text on blue circles. Image taken in Google earth V 9.123.0.2 (July 2019). Wyoming, USA. 41°48'02" N, 109°35'36" W, eye altitude 2625 m. © Google Earth.


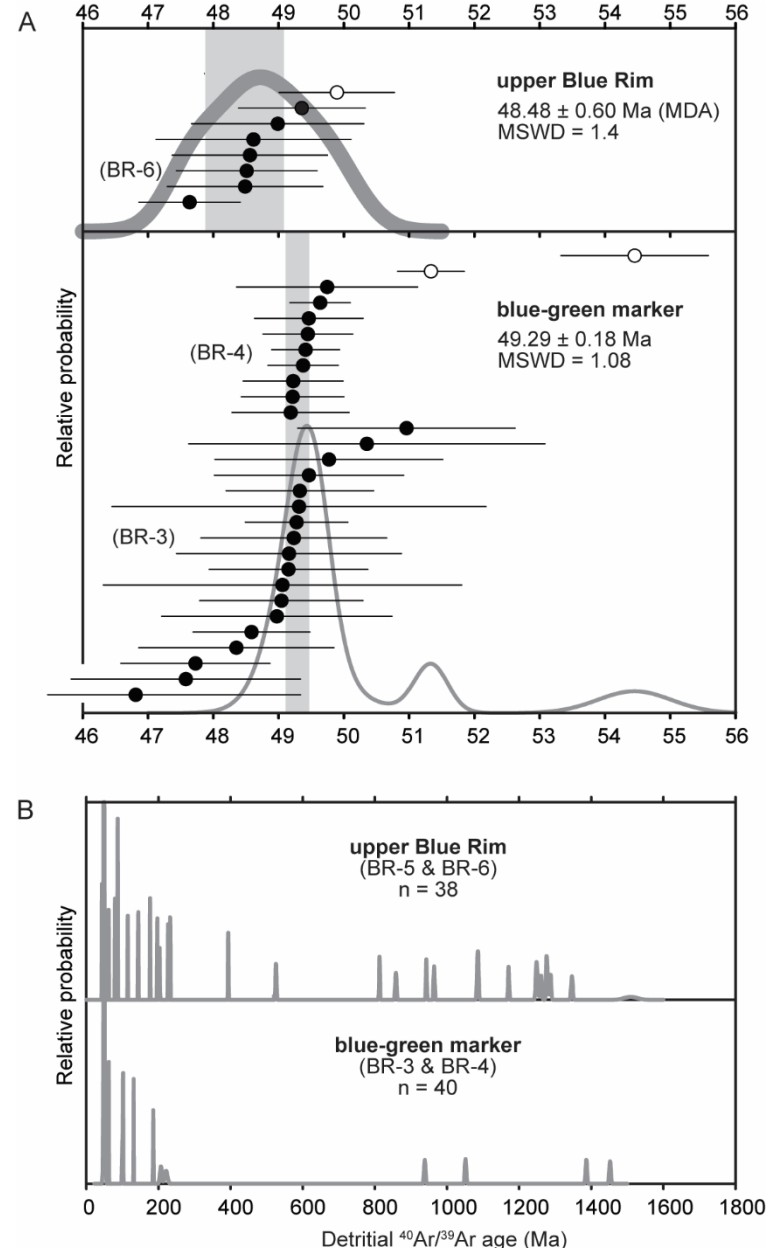

**Figure 8:** $^{40}$Ar/$^{39}$Ar geochronology: a) Relative probability plots of Eocene-aged sanidine from the volcaniclastic-lacustrine blue-green marker and an overlying pumice-bearing volcaniclastic sandstone (sample BR-6); b) Relative probability plot of $^{40}$Ar/$^{39}$Ar ages for detrital feldspar grains from the middle and upper Blue Rim, showing Phanerozoic and late Paleozoic ages characteristic of the Idaho paleoriver (cf. Chetel et al., 2011).

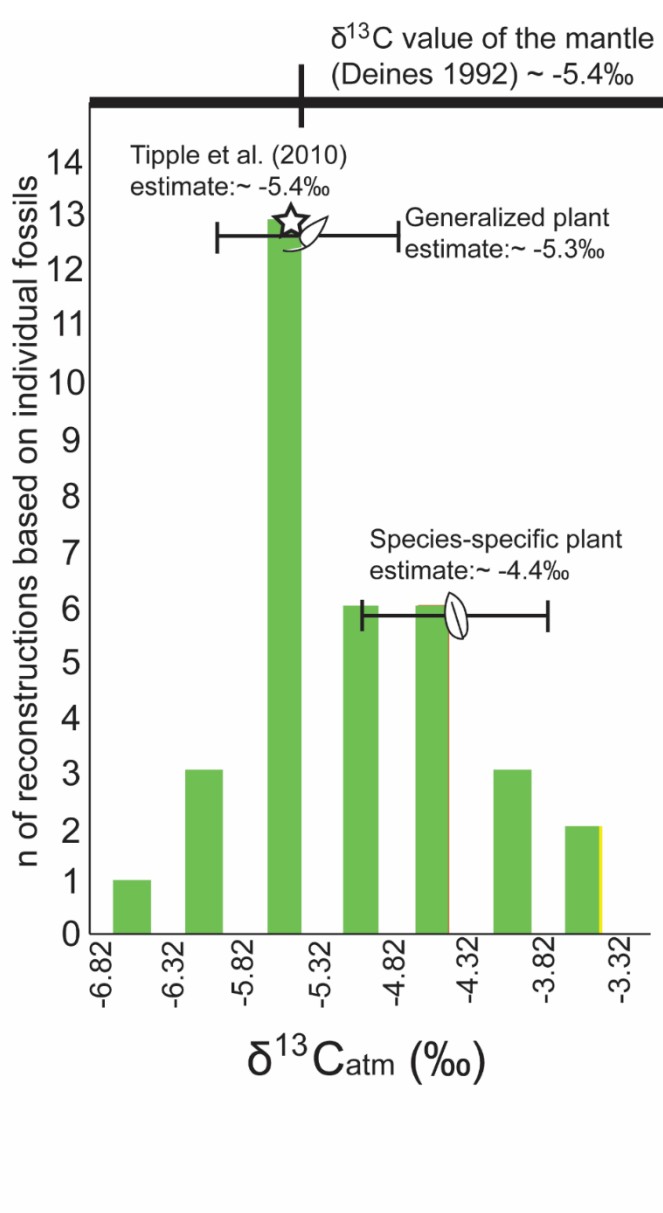

**Figure 9: $\delta^{13}C_{atm}$ as reconstructed from $\delta^{13}C_{leaf}$ of the Blue Rim fossil flora.** Reconstructions utilized the generalized relationship between $\delta^{13}C_{atm}$ and $\delta^{13}C_{plant}$ (Arens et al. 2000; $n = 34$; Equation 8). The species-specific mean and standard deviation are shown (based on *Lygodium* and *Populus* fossils, $n = 8$), as are the generalized plant mean and standard deviation. The Tipple et al. (2010) foraminiferal reconstruction is denoted in a star, and the $\delta^{13}C$ value of the mantle is shown above (Deines 1992).


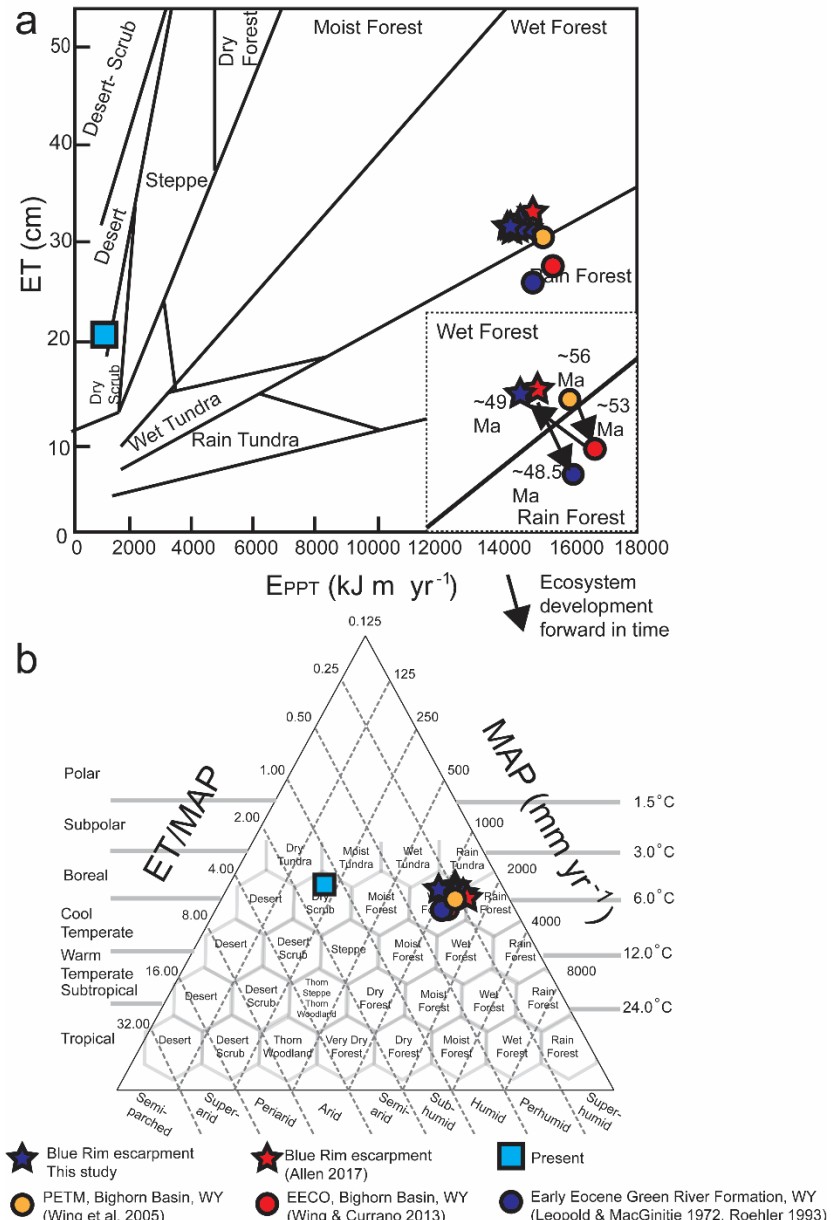

**Figure 10: Ecosystem level characterization of Blue Rim escarpment and other Cenozoic Wyoming ecosystems.**
Paleosol-based (a) Floral Humidity Province and (b) Holdridge life zones (Holdridge 1967) in blue stars. (a) includes inset showing climate progression over time, with (1) showing the PETM (Bighorn Basin, ~56 Ma), (2) the EECO (Bighorn Basin, ~53 Ma), (3) this site (~49 Ma) and (4) the latest stage of Lake Gosiute (Green River Formation, ~48.5 Ma; Smith et al., 2008). Comparative studies for this region based on nearby temperature and precipitation reconstructions are shown in yellow, red, and blue circles, while the climate of the present (as measured in Rock Springs, Wyoming, U.S.A.) is shown in light blue squares. Average values from Allen (2017b) using leaf margin analysis to reconstruct MAAT and leaf area analysis to reconstruct MAP is shown in red stars.

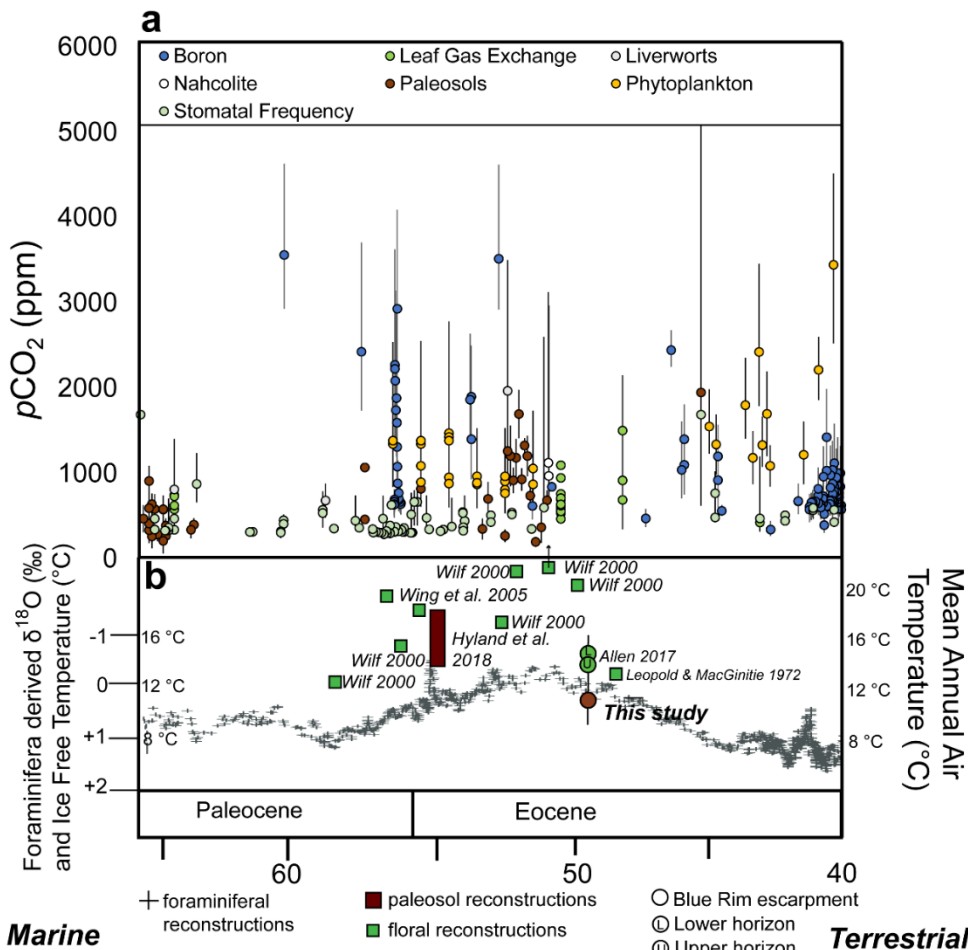

**Figure 11:** Paleoclimate reconstructions from Blue Rim escarpment in the context of the evolution of climate from 66 to 40 Ma. (a) $pCO_2$ from 66 to 40 Ma based on multiple proxies compiled on paleo-CO2.org (McElwain 1998; Ekart et al., 1999; Pearson & Palmer 2000; Royer et al., 2001; Greenwood et al., 2003; Royer 2003; Pagani et al., 2005; Lowenstein & Demicco 2006; Fletcher et al., 2008; Zachos et al., 2008; Franks & Beerling 2009; Retallack 2009; Bijl et al., 2010; Smith et al., 2010; Grein et al., 2011; Pagani et al., 2011; Hyland & Sheldon 2013; Hyland et al., 2013; Huang et al., 2013; Franks et al., 2014; Maxbauer et al., 2014; Jagniecki et al., 2015; Anagnostou et al., 2016; Barclay & Wing 2016; Liu et al., 2016; Kowalczyk et al., 2018; Witkowski et al., 2018; Zhang et al., 2018; Milligan et al., 2019; Steinthorsdottir et al., 2019; Haynes et al., 2020; Henehan et al., 2019; Henehan et al., 2020). (b) marine $\delta^{18}O$ and ice-free temperature reconstructions based on foraminiferal data (Zachos et al., 2008) and mean annual air temperatures reconstructed from physiognomic (green) and paleosol (brown) proxies in the Paleocene and Eocene. Error bars are from the paleo-CO2.org site and associated citations, and how they were calculated can be found in these sources.

**Table 1. Summary of single crystal sanidine $^{40}Ar/^{39}Ar$ analyses: Bridger Formation, Blue Rim, WY.**

| (sample) material | Latitude | Longitude | n | MSWD | Weighted mean age (Ma) | ± 2σ [†] | ± 2σ [‡] |
|---|---|---|---|---|---|---|---|
| **upper Blue Rim** | | | | | | | |
| (BR-6) sandstone | 41.8213° N | 109.5949° W | 6 of 18 | 0.93 | **48.29** | **± 0.45** | **± 0.48** |
| (BR-5) sandstone | 41.8210° N | 109.5952° W | 0 of 19 | detrital | n.a. | | |
| **blue-green marker** | | | | | | | |
| (BR-4) pumiceous sandstone | 41.8218° N | 109.5972° W | 9 of 20 | 0.24 | 49.43 | ± 0.23 | ± 0.28 |
| (BR-3) basal sandstone | 41.7987° N | 109.5834° W | 18 of 20 | 1.20 | 48.98 | ± 0.38 | ± 0.41 |
| Blue-green marker composite | | | 27 of 40 | 1.08 | **49.29** | **± 0.18** | **± 0.24** |

*Notes:* All ages calculated relative to the 28.201 Ma age for FCs using the equations of Kuiper et al. (2008) and Renne et al. (1998), using the decay constants of Min et al. (2000), and are shown with 2σ analytical and fully propagated uncertainties, which incorporate decay constant and intercalibration uncertainties. Neutron flux monitor: FCs-Fish Canyon Tuff sanidine, Cf. Table S2 for analytical details. Ages in bold reflect interpreted depositional ages.

[†]Analytical uncertainty.
[‡]Fully propagated uncertainty.