# Peer review of "Climate & Ecology in the Rocky Mountain Interior After the Early Eocene Climatic Optimum"

_Climate of the Past, 2021_

## Author Response (AR1)

**Climate of the Past, Article cp-2021-45**

**Point-by-point response**

*Reviewer comments are in unformatted regular text, while author responses are in bold.

Response to editor

Dear Dr. Stein,
Thank you for responding to the comments of the two critical reviewers. Both reviewers were positive about the overall direction and significance of the manuscript. Please submit a revised version of your manuscript that addresses the points covered in your response, paying careful attention to Reviewer 1's comments about uncertainty propagation and data quality for the bulk geochemistry data. I'm indicating "major revisions" in order to reserve the potential to send the revised manuscript for re-assessment by one of the original reviewers.
Sincerely,
Alberto Reyes

**We thank the editor for the positive feedback. We have made sure to address Reviewer 1's comments about uncertainty propagation and bulk geochemistry data, detailed below.**

Review #1

Title: Climate & Ecology in the Rocky Mountain Interior After the Early Eocene Climatic Optimum

Author(s): Rebekah A. Stein et al.

**General Comments:**

The study is interesting and addresses important scientific questions surrounding global versus regional environmental responses to past warm intervals and intervals of abrupt climate change. It is certainly relevant to and deserving of publication in Climate of the Past. This work contributes to the greater understanding of the North American terrestrial environmental response to carbon emissions, a timely topic when observations of modern shifts in the hydrologic cycle are considered. Further, it provides (1) new early Eocene proxy-based quantitative environmental constraints, and (2) new age constraints in a geologically significant area. The authors do a good job introducing the geologic setting and explaining their approach for environmental reconstructions and geochronology. The paper is also fairly well-structured and laid out in general. The explanation of various weathering indices was particularly well-written and concise. However, generally, the manuscript is only moderately well written, and would improve greatly with grammatical and sentence structure revision. Some of the arguments leading to main conclusions about atmospheric carbon sources in the early Eocene and Paleogene are weak or non-existent.

Generally, the majority of my critique involves the following:

- The study motivation and significance could be more clearly and effectively communicated. I give specific details below.

**We have added sentences to emphasize the motivation and significance throughout, including the suggested locations.**

- Information on the approach to analyzing for bulk geochemistry is opaque and needs to be expanded.
- Sentence structure and grammar needs to be improved. I aimed to give thorough and specific recommendations.

**We appreciate these suggestions and have made modifications.**

- Propagation of uncertainty and specifics on reported precision needs to be addressed, or at least better defined throughout the manuscript, with respect to paleo reconstructions.
- Discussion of environmental results and structure of discussion could improve.
- Some arguments leading to major conclusions are incomplete.

As stated above, this article is suitable for Climate of the Past and will be of interest to readers as it provides new paleoenvironmental constraints on an important interval. Based on the above critique, and the lack of thorough revision prior to submission, I recommend this article is reconsidered following major revisions.

**Specific Comments:**

-The study motivation could be improved or expanded upon. For example, the authors state how this warm interval may prove useful as an analogue for modern climate change for several reasons, but give the reader a weak connection between modern and past warming at that location using inconsistent plant fossils and hydrologic cycle comparisons. The reader is left wondering: "Why was it wetter then even though it was warmer and it's drying out now?", but the study doesn't specifically address this question.

**We have added text to clarify the connection, rather than asking audience to read between the lines, e.g., lines 61-82 "From the Paleocene to early Eocene, it has been inferred that there were extensive temperate forests dispersed throughout North America (Smith et al., 2012; Breedlovestrout et al., 2013; Greenwood et al., 2016; West et al., 2020) up to high latitudes 65 °N (Dillhoff et al., 2013). However, the nearby Bighorn Basin is inferred to have undergone aridification based on magnetic properties in paleosols (Maxbauer et al. 2016; Carmichael et al. 2017), and global climate models predict low and lower-middle latitude sites, including areas like central Utah to experience aridification due to changes in meridional vapor transport distribution (Pagani et al., 2006). As the planet warms, there is increasing concern about water availability and dry climates getting drier. For example, the North American Southwest, composed of a series of deserts and dry ecosystems, is at risk for having its already severe droughts increased in frequency and severity (Poore et al., 2005; Coats et al., 2015; Cheeseman 2016). Therefore, study of ancient climate and ecosystems in these hydrologically vulnerable areas can provide examples for what may happen to these ecosystems in the context of emerging climate and societal challenges.**

The connection between understanding this particular environment/location at this specific time and its significance to modern change is vague (especially with respect to the concluding sentence of section 1.1). The authors could build a stronger argument for study significance by stating that their study fits in a greater framework of understanding the global versus regional responses to carbon emissions and subsequent climate change, particular with respect to a shifting hydrologic cycle (i.e., observations of modern shifts in N. America hydrologic cycle can be better understood if given paleo-context). Further, this region and the Cenozoic sediments it contains are well-studied. In the introduction, the manuscript would benefit from a more thorough explanation of the significance of this study with respect to previous work and understanding of the region. The authors do a good job of contextualizing this data in the discussion (section 5.1). However, this should also be laid out as a study motivator in the introduction, not just the paleo analogue argument, in my opinion.

**Thank you for this suggestion. We have clarified our motivations in the introduction, finishing section 1.1 "Observations of modern shifts in the North American Southwest hydroclimate can be better informed with a paleo-context, and as such, we focus on paleohydroclimate changes in this region, contextualized with similar regional studies from this time throughout the Rocky Mountain region (e.g., Leopold & MacGinitie 1972; Wing & Greenwood 1993; Greenwood & Wing 1995; Inglis et al., 2017; Murphey et al., 2017; Allen 2017a/b). This study fits in a greater framework for understanding global and regional responses of terrestrial climate, and more particularly, terrestrial hydroclimate, to carbon emissions."**

-Line 45: Cite refences here which constrain the interval of warming you state. I recommend looking into Westerhold et al., 2018 or Cramwinckel et al., 2018.

**Thank you for the suggestion, we have added both reference suggestions.**

-Line 170: The elements analyzed should be listed in this section.

**Added, thank you for the suggestion.**

-Line 173: Please explain what you are using for "internal standards." Is this an in-house multi-element solution standard at ALS? Also, how is precision defined here? How is it determined? For example, is it determined using 2SE of long-term reproducibility in solution consistency standards? Or, perhaps, 1sd of multiple measurements of an individual sample across many analytical sessions? Generally, this section needs some more details for the ICP-informed reader.

**Samples were analyzed in a commercial laboratory (ALS National Laboratories in Vancouver, BC) according to their proprietary methods. We do know that error was calculated based on the maximum error of duplicate and standard tolerance, but that is the extent of public-available information. When contacted, they did not provide further information.**

-Line 206-208: I cannot make sense of this sentence. It needs revising. Certainly, consider removing the word "so" and/or state "U/Th is redox-sensitive" parenthetically rather than in commas.

**Corrected, thank you.**

-Line 315: You should refer readers here to the sedimentary geochemistry data which you make available.

**We added the data repository at the start of this section, thank you for this suggestion. Mendeley Data Repository, Doi: 10.17632/z6twpstz4r.3**

-Line 337: How is the precision in temperature reported? Does this standard deviation you report consider the analytical uncertainty in bulk geochemistry used in the PWI calculation (i.e., 0.2 wt %)? Does it include calibration uncertainty in the equation which translates PWI to temperature? Is it simply based on temperature reproducibility (i.e., the standard deviation of multiple sample temperature values) with no propagation of analytical or calibration error? Sorry for all of the questions, but it is important to be transparent here. If PWI-temperature calibration uncertainty was not previously constrained, it may be best to give the reader an estimate of the fit of the calibration regression by providing an $R^2$ value from Gallagher and Sheldon, 2013. If uncertainty is constrained in this relationship, please utilize it by propagating into the temperature uncertainty and state that you are doing so.

**Clarified in text. The standard deviation does not include the uncertainty in bulk geochemistry used in PWI. The standard deviation just looks at multiple sample temperature values, and is based on reproducibility. We then compare it to the calibration uncertainty translating PWI to temperature, and demonstrate that the standard deviation is within error of calibration uncertainty.**

**The standard deviation on replicate analyses on six profiles from the same paleosol was smaller than the error on the proxy itself. The analytical uncertainty on PWI is that of ALS laboratories (see above), so the error is dominated by error in the model calibration.**

-In figures where error bars are being used, more details with respect to error propagation is needed similar to the critique above on bulk geochemistry reproducibility.

**Figure captions have been updated appropriately.**

-Line 384: I don't think you can say "slightly" here given your MAAT of 11ºC from a paleolat in the low 40ºs N, and the 35ºC MAAT from 36º N latitude.

**Clarified in text, slightly to moderately depending on the latitude.**

-Lines 390-394: Here, seasonality in temperature is brought up, and despite that this is the discussion, no discussion on potential cold or warm season biases in the authors' temperature reconstructions are brought up. This is necessary.

**Paleosol B-horizon elemental chemistry should not have a seasonal bias because it takes so long to form (typically hundreds to thousands of years. Given that length of time, B-horizons are in equilibrium with the environment and integrate the environment over time. One of the pros, and the cons, depending on the resolution that you seek, is that elemental chemistry is unable to reflect seasonality. Of note, clumped isotopes in carbonates (and carbonate nodules in general) do have a seasonal bias related to time of year they form dependent on timing of precipitation and temperature. We made this explicit in the text.**

-Line 396: What paleosol-based results? All of them or just the temperatures? Confusing as written.

**Clarified to include temperature and precipitation.**

-Line 399: Your temperature results are similar to Wing et al. (2005)? It does not appear so to me. To which PETM data in Wing do you refer: min body CIE temps or pre-/post-event temps? Surely your data can't be similar to both considering the warming at the PETM and your reported precision… that is, if temperature is what is being discussed here.

**We meant the ecosystems present were comparable, which we clarified in text.**

-Line 408: Yes, because they are within error, but also because of data scarcity and sampling frequency, no?

**Based on Dzombak et al. (2021) in Palaeo-3, we found that paleosol-based reconstructions based on the number of sampled paleosols (n = 6) was sufficient to minimize error (see Dzombak et al., in press, Palaeogeography, Palaeoecology, Palaeoclimatology; https://www.sciencedirect.com/science/article/pii/S0031018221004260).**

-Line 412-415: Please expand with citations. How does a past warmer climate allow for inceptisols to form in warmer conditions? Without details or a mechanism, this comes across circular and non-scientific.

**We agree and have removed this text.**

-Lines 423-424: As written this is a bold statement given the dataset. How are you sure that you simply didn't sample shorter-term climate variability? This needs a timescale associated with it, such as: "climate was likely generally steady (+/− < 5ºC) on 100kyr+ timescales."

**Clarified in text, i.e., line 584: "we interpret the overall climate as relatively steady (± 5 °C) on 100,000 year or more timescales during this interval."**

-Line 426-427: There is not much of a debate if you only include one reference here. This statement falls within the realm of marine work. The most reliable reconstructions of the early Eocene in terms of temp and pCO$_2$ are from marine archives and they should be cited and discussed here (e.g., Anagnostou et al., 2017; Cramwinckel et al., 2018).

**We included these additional references.**

-Line 429: The methane release hypothesis needs a citation (probably a Jerry Dickens paper), and volcanism could use a few other citations (e.g., Gutjahr et al., 2017 and a recent article constraining the magnitude of North Atlantic Igneous Province volcanism).

**Added additional references (Dickens 2011; Gutjahr et al., 2017, Jones et al., 2019).**

-Line 432-434: Unfortunately, as written this is incorrect and a very surficial explanation of the complexity of the scientific problem at hand. What you state about reconstructing the C source using $\delta^{13}$C is not possible without an additional constraint on parameters such another constraint on surficial carbon cycling (e.g., CCD) or temperature + climate sensitivity.

**One of the advantages of using a plant-based reconstruction technique is that it does not rely on things like the CCD because plants live in direct contact with the atmosphere. We agree that there are complexities in using fossil forams to reconstruct the atmosphere. We find it heartening that**

**this technique, which is independent/separate from ocean chemistry, finds similar answers to reconstructions based on ocean chemistry. While we agree that it is not possible to fully reconstruct all sources to the atmosphere without more information, we disagree that we have over-interpreted. As they stand, the discussion of atmospheric $CO_2$ sources states that this study provides additional evidence that the source had an isotopic value of the mantle.**

-Line 445-446: Please explain how $\delta^{13}C_a$ of ~−5.3 to ~−5.8 provides evidence that increases in atmospheric $pCO_2$ over the LPEE were driven by a volcanic source. Your data do not support this conclusion without other constraints on climate or the carbon cycle, and there is no clear argument provided in the text to support this conclusion. In addition to an atmospheric $\delta^{13}C$ value, one must understand and constrain the global exogenic carbon cycle to know the long-term driver. If you are arguing that (1) your values are similar to what Tipple et al. (2010) came up with, and (2) That study claimed to constrain the driver of long-term $pCO_2$ increases, thus your value supports that hypothesis, you are incorrect in your written statement and should remove this sentence. If this is not your intention, please more clearly explain why your new values help support this previous finding. Please also see Komar Zeebe and Dickens (2013) for a detailed study involving geochemical constraints on the long-term drivers of LPEE $pCO_2$ increase using C cycle box model.

**We have added this reference and included a sentence very explicitly mentioning these limitations.**

-Besides small local volcanics, if you state that your data supports a certain C source, you should point out and discuss (e.g., magnitude of C) the hypothesized source of volcanism for the Paleogene: North Atlantic Igneous Province Volcanism.

**We have noted the presence of the NAIP and added a citation.**

-Line 446-447: Citation for "period of elevated rate of volcanism" needed. This sentence states that global $CO_2$ and temperature drove a slowing of volcanism written as is. I don't think that is intended by the authors, and it should be revised. Also, the Zachos et al. (2008) citation is suboptimal and a more recent study which investigates the cause of EOT cooling should be utilized. Zachos et al. (2008) do not specifically point to a decrease in volcanism to be the driver of the EOT.

**We have clarified in text the period of elevated volcanism we meant.**

**Technical Corrections:**

-Line 11: Confusing/redundant to say that increasing temperatures "accompany" modern climate change. Consider revising.

**Done.**

-Line 14: Here you spell "analog" and below in section title 1.1 you spell "Analogue."

**Done.**

-Line 24: "at that time" is confusing as it refers to when you went about reconstructing environmental conditions written as is. Consider removing phrase.

**Done.**

-Line 35 and throughout: You are using hyphens (-) instead of negative signs (−).

**Thank you for pointing this out. We have addressed this.**

-Line 179: Equation numbers appear misaligned with those below (possible formatting issue).

**Done.**

-Line 192: Above there is an extra line spacing after equations; it is missing here.

**Done.**

-Line 201: "Was" should be "were."

**Done.**

-Line 203: "The molar ratio" should be "The molar ratios."

**Done.**

-Line 220: Citation "2017b" with no author. Double check this is the appropriate format for CotP. I am uncertain since the text mentions the coauthor by initials.

**Done.**

-Line 250: New paragraph needs indentation.

**Done.**

-Line 289: "Figs. 4-7" Use em dash instead of hyphen.

**Done.**

-Line 290: Remove "anywhere."

**Done.**

-Line 295: "Inceptisols" paleosol capitalized throughout. I do not think this is common practice, but I could be wrong.

**USDA soil orders are considered proper nouns and are capitalized.**

-Line 302: Change to: …typical of values… or revise sentence.

**Done.**

-Line 305: "Demonstrated" confusing. Consider changing to "Displayed?"

**Done**

-Line 318: Here you are using "percent C" and "percent N", but above they were "%C" and "%N". Reminder to keep things consistent.

**Done.**

-Line 322: "This specific field excursion (2019)" is a bit confusing. Consider rewording to "the 2019 field excursion."

**Done.**

-Line 329: Missing word. "located at" or similar instead of "located. "

**Done.**

-Line 344-347: Extra word: "are", and many other confusing errors with this sentence. Requires revision.

**Done.**

-Line 365: "which can be interpreted to mean that" can be more concise. For example, "which may suggest".

**Done.**

-Line 371: Remove "actually" (informal/needless).

**Done.**

-Line 372: Vague. How are they consistent? Consider rewording sentence to state that "Changes in X element ratios are consistent with…"

**Done.**

-Line 384-385: Confusing, grammatically incorrect sentence.

**Done.**

-Line 385-387: State that this is the range in temperatures for the early Eocene (correct?).

**Done.**

-Line 399: Capitalize "Thermal" and "Maximum."

**Done.**

-Line 410: I don't think it's common practice to capitalize these paleosol names.

**It is common practice to capitalize any paleosol name that overlaps with modern USDA taxonomy.**

-Line 421: As written, this reads as if the "discrepancy" "represents modest actual change…" rather than the data/reconstruction.

**Done.**

-Line 432: "…processes and landscapes" change to "…processes and landscapes to be mobilized into the atmosphere" or similar. As is, this sentence is unclear.

**Clarified.**

Review #2

Manuscript: Climate & Ecology in the Rocky Mountain Interior After the Early Eocene Climatic Optimum

Authors: Stein et al.

Journal: Climate of the Past

Reviewer: Erik L Gulbranson

**Overview**:

This study reports on an ensemble of paleosol-based paleoclimate proxies and new geochronologic dates of Eocene strata of Wyoming. The use of multiple proxies and critical assessment of the performance of these proxies through comparison to leaf physiognomy and isotope proxies for paleoclimate is a robust contribution towards methodology and an understanding of the Early Eocene Climatic Optimum. The authors advance a proposal that volcanic degassing of $CO_2$ likely contributed to the sustained warming during EECO based on carbon isotopes of foliar material collected in these strata.

**We appreciate the reviewer's positive feedback.**

The stable carbon isotope analysis is likely the weakest part of this study, however, I do not think this is a fatal flaw. Tempering of the significance of the implications of the stable carbon isotope results will help this manuscript achieve its greatest impact without sacrificing credibility. I recommend enhancing the focus of the stable carbon isotopes on species specific trends (if they exist) among the studied taxa.

We thank the reviewer for this positive feedback. We have tempered the significance of the implications to appropriately reflect our confidence, and emphasize that this is useful only in the context of marine proxy reconstructions. We unfortunately did not have a high enough density of specific species to make reconstructions based on these, though we do agree that would be ideal.

The writing can be improved substantially in several sections of this manuscript, and in some particular cases avoid unnecessary confusion or distraction from key aspects of this study.

**We appreciate the reviewer's comments and have made modifications throughout to streamline the message.**

**Major-level comments:**

None.

**Moderate-level comments:**

Redrafting of several sections in this manuscript is advised to increase the clarity of the writing and strength of the arguments presented.

**Thank you for this suggestion. We have clarified the unclear sections, as documented below. Some of this included changing or defining word choice, i.e., paratropical. Some of this included contextualizing the broader argument, i.e., CO2 isotope systems, timescales, details on paleosols. More details can be found below.**

Paleosol descriptions are lacking.

**We agree that more details are needed, and because most of the readers will not be paleosol specialists we have added a Supplemental Table S3 including our detailed paleosol observations.**

Reliance on Arens et al., 2002 may be a critical weakness in the security of the conclusions on volcanic degassing.

**Yes, we agree that the transfer function is not particularly precise. However, this is why we ran 34 analyses. The 34 analyses mean that the uncertainty on the reconstructed atmospheric value is much smaller than anything based on the proxy by itself.**

**Minor comments:**

See below.

**Line-by-line comments:**

Line 22: Given how this reads it is inaccurate to say that provenance and parent material was "reconstructed". Neither of these variables has been reconstructed (i.e., if I reconstructed provenance of a sediment, then I would attempt to create or synthesize an erosion, transport, and deposition scenario that mimics what is seen in the rock record), but they have been studied to identify the source of sediment and composition of the parent material in these Eocene strata.

**We have replaced this word with studied. Thank you.**

Line 24: There are two possible isotopic systems in $CO_2$. Be specific about which system, carbon in this case, is being calculated via proxy.

**Clarified.**

Line 25: This sentence should be broken up into two sentences with the second sentence discussing the comparison.

**Done.**

Line 28: Comparing paleosol to foliar-based paleoclimate proxies makes me think of time-averaging (irrespective of the uncertainty in each proxy). How comparable are foliar-based paleoclimate proxies to paleosol paleoclimate proxies if the paleosol represents 100 years, 1000 years, 10000 years, etc.? Moreover, at this early stage of the paper I'm also wondering if these paleosols may be polygenetic and thus integrate geochemical archives of different climate states, or are these solitary profiles where we can be certain that the paleosol developed in equilibrium with the state factors at that time? I'm interested to learn more about this in this paper, but this is an opportunity to clarify these issues for the reader in the Abstract.

**Clarified in text. There are no thousand year old leaves on trees, so time-averaging on trees is irrelevant. There are many angiosperms in the assemblage, which drop annually, but the section has several quarries of amassed flora, likely controlled by preservation rather than actual shedding; thus this preservation is likely to span a larger period than the leaf shedding period.**

Line 30: It is apparent now that I'm not clear on what the purpose is of this study, the problem or hypothesis that was to be tested or evaluated. I re-read the earlier parts of the abstract to see if I missed something, but the purpose of this study I think is more implied than a direct statement. Please consider revising the first 1/3 of the Abstract to better elucidate this.

**We have added a sentence before launching into the details of the study "Using this well-preserved basin deposited during a period of tectonic and paleoclimatic interest, we employ multiple proxies to study trends in provenance, parent material, weathering and climate throughout one million years." on line 18. This is after the background about why we care about the Eocene, but before the logistics of the study.**

Line 47: This is overly generalized and inaccurate. The PETM included pronounced regions of aridification and associated landscape, floristic, and vertebrate changes. This also establishes a contradiction with the next sentence, which also lacks crucial clarity as to the mechanism(s) for why an already dry climate may become drier under increasing atmospheric temperatures.

**Addressed by alluding to the complexities of this time period. This makes the transition to the concern about hydroclimate smoother.**

Line 50: There is more than one desert in this broad region. Is it true that all of these deserts are equally affected in terms of response, timing, and magnitude to a given climate forcing?

**Done.**

Line 59: What specific mechanism(s) led to the formation of a series of large lakes?

**Clarified in text, increased and changed fluvial flow due to uplift of mountains..**

Line 65: A point of clarity, the Laramide structures probably did not contribute water to anything at the Earth surface, rather (and I'm assuming the original meaning), as uplifted blocks they may have influenced the transport of atmospheric moisture and groundwater flow paths in the region.

**Clarified in text to state that they influenced atmospheric transport.**

Line 69: This is a very precise paleolatitude, 41.82ºN, what is the uncertainty on this estimate? However, with more definite knowledge of the modern latitude of the region, the comparison should be more definite than "is thought…"

**The uncertainty exceeds the utility of having that precision, so we have corrected it to say ~41N.**

Line 70: This sentence can end after the word "latitude".

**Done, thank you.**

Lines 73–75: I understand the purpose of this introduction, but it requires some revision: 1) consistent format for references; 2) breaking the reference to specific proxies out of the parentheses and into a sentence or two; 3) describing more of the connection of an observation (e.g., isotope value) to an interpretation.

**Done.**

Line 75: How are the quality of organic specimens determined?

**Clarified.**

Lines 76–78: This is overly vague and lacks key references.

**Added references and clarified the importance of this statement.**

Lines 78–79: Again, overly vague. Could this section instead be rolled into the Methods section? In this version of the manuscript this section doesn't really add any information.

 **This section could be placed in the introduction or methods, depending on the desired message. In this case, we are emphasizing the importance of using multiple proxies to understand an environment. As such, we are leaving it where it is but changing the tone and focus.**

Line 96: What are the uncertainties on these ages, and have these ages been corrected so that they are comparable to U-Pb ages?

**Yes, all the ages we report or discuss have all been calculated using the 28.201 Ma age for Fish Canyon tuff sanidine standard (Kuiper et al., 2008). We actually did this in a 2010 paper, which we cite in the text:** *Radioisotopic ages reported or discussed in this contribution have all been calculated using the 28.201 Ma age for the Fish Canyon tuff sanidine standard, and are thus comparable with modern U-Pb geochronology (Kuiper et al., 2008; Smith et al., 2010).*

Line 98: Siliciclastic is a general term that implies a quartz-rich clastic sediment. When I read about potential sediment source areas I am generally surprised to see siliciclastic as the first potential source listed without specific mention of sediment recycling. Regarding provenance I think it makes sense to start with the most fundamental data available, sediment composition, and then work backwards to identify probable sources of that sediment.

**These potential sources exist and have been characterized previously. There isn't an explicit sandstone petrography dataset being interpreted here. Some more detail on the siliciclastic sediments include: a) the most common detrital feldspar ages in the sand are similar to depositional age (i.e., recently erupted volcanic grains) and b) there are lots of euhedral volcanic biotite and felsic volcanic lithic grains (small pumice clasts) in Bridger Fm sandstones (Chetel et al., 2011). Additional characterizations are included in Smith et al., 2008; Smith et al. 2015:**

**Smith, M. E., Carroll, A. R., and Mueller, E. R., 2008, Elevated weathering rates in the Rocky Mountains during the Early Eocene Climatic Optimum: Nature-Geoscience, v. 1, p. 370-374.**

**Smith M.E., Carroll A.R., Scott J.J. (2015) Stratigraphic Expression of Climate, Tectonism, and Geomorphic Forcing in an Underfilled Lake Basin: Wilkins Peak Member of the Green River Formation. In: Smith M., Carroll A. (eds) Stratigraphy and Paleolimnology of the Green River Formation, Western USA. Syntheses in Limnogeology, vol 1. Springer, Dordrecht. https://doi.org/10.1007/978-94-017-9906-5_4**

Line 109: What about the roots, and anatomical attachment? I would temper this statement to refer to excellent plant fossil preservation without the qualifier of "all plant organs".

**Done.**

Line 110: This is broader critique I have with paleobotanical references to biomes in general. Given the paleolatitude being in the mid-latitude region, this cannot be a subtropical biome, sensu stricto. Rather, the flora contained in this biome may contain elements consistent with biomes at lower paleolatitudes, which says something important about the Etp/MAP balance, seasonality, MAT, etc. What it doesn't say, which is what subtropical suggests, is that the incident angle of solar radiation was the same at ~41ºN as it is between 0º and ~25º, and remains with a finite difference through one full rotation around the Sun. It also doesn't say that Hadley Cell circulation was different, where the descending limb extends to ~50ºN, which is what is implied by calling this biome subtropical. Instead, we have a mid-latitude region, with mid-latitude sunlight seasonality and power, with mid-latitude atmospheric circulation (or lackthereof), with a flora that previously (and afterwards) inhabited only the equatorial latitudes. Personally, I think this showcases the significance of this latitudinal shift in flora and points to some of the key aspects to study the who/how/and why about how these ecosystems came to develop here. If the mid-latitudes truly became subtropical in every sense of the word, then that would be likewise astounding, but is that what we're saying here?

**This is an interesting and helpful point. Rather than saying we found subtropical ecosystems at 41N, we have corrected the text to say ecosystems comparable to modern subtropical ecosystems, an important distinction!**

Line 112: We don't know that there are quarries or what these are quarries of, this sentence needs a segue of some sort.

**Clarified.**

Line 121: How do the authors know that these are volcaniclastic beds?

**A couple of reasons: a) the most common detrital feldspar ages in the sand are similar to depositional age (i.e., recently erupted volcanic grains) and b) there are lots of euhedral volcanic biotite and felsic volcanic lithic grains (small pumice clasts) in Bridger Fm sandstones (Chetel. Et al. 2011).**

Line 122: The blue-green marker needs a more precise and archivable definition. How would a person unaffiliated with this research team find this bed? I see more specifics later on, but at this juncture there should be a reference to a figure or something to direct our attention to where this bed is.

**Done.**

Line 139: What is meant by "updated stratigraphic column"?

**Done.**

Line 140: Arbitrary sampling is fine, but what was the rationale for this choice in sampling?

**This gives us reasonable statistical coverage relative to the thickness of the section. We were not looking for a discrete event, so we did not need higher resolution sampling than this.**

Lines 152–153: Epipedons are preserved in all of these paleosols?

 **Clarified.**

Line 165: Which references were used for the C isotope analysis and what was the performance of those references on the Picarro over the time range of the analysis of these samples?

 **Clarified.**

Line 210: Please change soils to paleosols (admittedly, I refer to paleosols as soils all the time, but it is not appropriate).

 **Done.**

Line 219–220: It would be more concise to cite this dissertation along with a description of the method used.

 **The geochronologists on this paper said this was the community standard.**

Line 240: It's hard to quote that $R^2$ and be confident in the results. However, what really crushes my confidence in the approach of Arens et al. (and helps explain the low $R^2$) is the fact that they hold constant the variable that plants modulate to respond to climate (Ci/Ca). It's just not a sound approach. I know it is used widely, but, that's just not a sufficient reason for me to agree with it. Paleoclimate is the goal for many of us, but it's how we arrive at our conclusion that matters. As an analogy, I can measure the stable isotopes of carbon in practically any carbon-bearing substrate. Those techniques are pretty easy, but if my data is to mean something I need to carefully select my samples and process them in such a way to preserve the signal I hope to extract from these samples.

**We agree with the reviewer that the $R^2$ value in the Arens et al. (2000) model is low, though significant. Of note, we agree with the reviewer and Arens et al. that Ci/Ca is likely not variable, despite being held constant for this model. However, the complexity of this measurement is not yet constrainable in geologic time. As such, we used a large number of measurements to assess the *most likely* $\delta^{13}C_{atm}$ value for comparison to marine d13Catm reconstructions. We would not interpret any of the individual measurements as reasonable, however based on the clustering of the data, the uncertainty on the reconstructed atmospheric value is much smaller than anything based on the proxy by itself. We are heartened by the comparable marine-based reconstructions.**

Section 4.2 Where are the O horizons? For paleosol 1, it is missing a B horizon, but does this mean that it has an A horizon over a C horizon, where the A and C horizons are separated by an erosional contact? If so, then there are many possibilities for what that profile may represent, but, it wouldn't represent a continuum of soil-forming processes. For paleosol 4, I highly doubt that erosion of the A horizon took place during the burial process, which as the name implies, indicates burial of the strata. It is more often than not the case that the epipedon of paleosol profiles are removed via erosion when those profiles are formed in overbank regions of fluvial environments or proximal to shorelines of

lakes/shorefaces. After that erosion, and subsequent deposition of new material the profile may be buried, preserving its truncated form. This sections needs a systematic description of the paleosol profiles, followed by their diagnosis against your taxonomic scheme of choice.

**We have added Supplemental Table S3 with details on the paleosols. All questions should be addressed there.**

Line 295: Without a systematic presentation of the paleosol observations and their lateral variation it is not clear how these profiles represent Inceptisols rather than Entisols. Also, the Soil Survey Staff, 2014 should be cited here.

**See Table S3 for details, and Soil Survey Staff citation has been added.**

Line 335: What was the % difference in CIA-K in the A or B horizons relative to the parent material? I use an arbitrary cutoff of 5%, with the idea that the greater the difference of the subsoil relative to the parent material indicates a greater likelihood that the soil formed closer to equilibrium with its environment (and thus that the solid state major element concentrations reflect all of those lovely contributions of weathering energies from water and organic acids).

**The two paleosols excluded from the analysis (19BRWY1 and 2) did not show CIA-K values >5% of the parent material, but the remaining ones, 19BRWY3-19BRWY6, had values >9% throughout, indicating they were in equilibrium with their environment.**

Line 340: Phew, I was really hoping to read this statement (species-specific tests). What I mean is, the authors have identified these plant taxa with wide ranging ecology, and it would be expected/anticipated to see carbon isotope variation among them (maybe clueing us into ecosystem processes as a function of functional diversity). I'm excited to read more.

**Thanks!**

Line 367: What were these oxygen isotopes measured on?

**Micritic lacustrine carbonates. We have added this, thank you!**

Line 375: This makes sense as the name Blue-green marker bed suggests a sedimentary unit with either stratified or variegated color of blue-green, which is indicative of reducing conditions.

**Thank you for this comment. We have added that clarification.**

Line 379: $R^2$=0.2 suggests that this explanation does not satisfactorily explain the variance in the data. Moreover, this explanation is fairly weakly held as a taphonomic difference could also explain the high/low carbon content. If the authors wish to further test this, then a compound-specific analysis (maybe via pyr-gc or solid-state 13C NMR) could be informative on the composition of the organic carbon (granted you'd be looking for the diagenetic products of specific ensembles of organic acids).

**This is a great point. We have changed the language to reflect that while this correlation is interesting (albeit weak), this could be related to taphonomic difference.**

Figure 3C: It is difficult to read the text superimposed on the image.

**Thank you, we have fixed this.**

Figure 5B: Why is the scale set to 0.002? This is an exaggerated scale when none of the data plot even half the way to this value.

**Set to a more reasonable value to show the very small amount of variability.**

Figure 7: A strike and dip symbol would be instructive on this image.

**We added a strike and dip symbol as well as the location of the stratigraphic column for context.**

Figure 9: Is the color spectrum just the same representation as the y-axis? If so, it is redundant, confusing, and should be discarded in a favor of a more simplistic visualization of this data (e.g., without color).

**We have simplified the figure, thank you to the reviewer for this feedback.**

---

## Author Response (AR2)

*Response to Reviewers*

*cp-2021-45 "Climate & Ecology in the Rocky Mountain Interior After the Early Eocene Climatic Optimum" Stein, Sheldon, Allen, Smith, Dzombak, Jicha*

General Comments
The authors did an excellent job of addressing reviews. I apologize for the delay in completing the follow-up review. The project motivation is much clearer following revisions, and the contextualization of their data has increased straightforwardness and readability for a broader paleoclimate audience. The study is interesting, providing an important snapshot of paleoenvironmental conditions during an interval of dynamic climate from a key locality. Overall, the paper is well-organized, well-written, and well-suited for publication in Climate of the Past in its current state. I have included some very minor comments, most of which will likely be addressed during the editorial process. These minor comments can be treated as suggestions; I am recommending publication as is.

**We thank the reviewer for their positive feedback.**

*Minor Comments*
Lines 385-388: Sentence is unclear.

**We have edited the sentence to be clearer: "One limitation on this reconstruction is that it does not account for species-specific isotope discrimination behaviour that varies taxonomically (Beerling & Royer 2002; Stein et al., 2019; Sheldon et al., 2020). Values on the fringes of these values (44%) may be so extreme after having experienced diagenesis of certain compounds while others were left behind, skewing the isotopic values to be representative of the compounds and not the bulk tissue (Beerling & Royer 2002; Tu et al., 2004)."**

Lines 428-429: Sentence is unclear.

**We have edited the sentence to be clearer: "More compound-specific analyses of organic compounds would be needed to determine if and how plant influence is contributing to weathering in high % C sections (Fig. S4; $R^2 = 0.20$; p-value = 0.01; Ong et al., 1970; Berner 1992)."**

Lines 457-460: Thanks for adding a bit more discussion on potential for seasonal biases. The last sentence of this paragraph would improve if a reference or two was included to support the phrase "B-horizon bulk geochemical data is in equilibrium with the environment and takes so long to form, therefore is not resolved enough to be affected by seasonality."

**We have added a reference (Sheldon et al., 2002). Thanks!**

Line 460-461. Sentence is unclear or incomplete.

**We have modified the sentence to be clearer. "These collected fossils (sampled at 26 m on the stratigraphic section, housed at ESS laboratory at the University of Michigan) are taxonomically comparable to fossils found by Allen (2017b; Bridger Formation) and MacGinitie (1969; Green River Formation). The flora sampled are characteristic of a mesic, forested environment (e.g., wet forest; Hamzeh & Dayanandan 2004; Hamzeh et al., 2006)."**